# Amortised Invariance Learning for Contrastive Self-Supervision

**Ruchika Chavhan**[1], **Jan Stuehmer**[2,3,*], **Calum Heggan**[1], **Mehrdad Yaghoobi**[1], **Timothy Hospedales**[1,4]
[1]University of Edinburgh, [2]Karlsruhe Institute of Technology, [3]Heidelberg Institute for Theoretical Studies
[4]Samsung AI Research Centre, Cambridge, *Work done while at Samsung AI.
Correspondence: `R.Chavhan@sms.ed.ac.uk`, `t.hospedales@ed.ac.uk`

## Abstract

Contrastive self-supervised learning methods famously produce high quality transferable representations by learning invariances to different data augmentations. Invariances established during pre-training can be interpreted as strong inductive biases. However these may or may not be helpful, depending on if they match the invariance requirements of downstream tasks or not. This has led to several attempts to learn task-specific invariances during pre-training, however, these methods are highly compute intensive and tedious to train. We introduce the notion of amortised invariance learning for contrastive self supervision. In the pre-training stage, we parameterize the feature extractor by differentiable invariance hyper-parameters that control the invariances encoded by the representation. Then, for any downstream task, both linear readout and task-specific invariance requirements can be efficiently and effectively learned by gradient-descent. We evaluate the notion of amortised invariances for contrastive learning over two different modalities: vision and audio, on two widely-used contrastive learning methods in vision: SimCLR and MoCo-v2 with popular architectures like ResNets and Vision Transformers, and SimCLR with ResNet-18 for audio. We show that our amortised features provide a reliable way to learn diverse downstream tasks with different invariance requirements, while using a single feature and avoiding task-specific pre-training. This provides an exciting perspective that opens up new horizons in the field of general purpose representation learning.

## 1 Introduction

Self-supervised learning has emerged as a driving force in representation learning, as it eliminates the dependency on data annotation and enables scaling up to larger datasets that tend to produce better representations (Ericsson et al., 2022b). Among the flavours of self-supervision, contrastive learning has been particularly successful in important application disciplines such as computer vision (Chen et al., 2020c; Caron et al., 2020; Zbontar et al., 2021), medical AI (Azizi et al., 2021; Krishnan et al., 2022), and audio processing (Al-Tahan & Mohsenzadeh, 2021). The key common element of various contrastive learning methods is training representations that are *invariant* to particular semantics-preserving input transformations (e.g., image blur, audio frequency masking) that are applied synthetically during training. Such invariances provide a strong inductive bias that can improve downstream learning speed, generalisation, and robustness (Geirhos et al., 2020).

A major vision motivating self-supervision research has been producing a general purpose representation can be learned once, albeit at substantial cost, and then cost-effectively re-used for different tasks of interest. Rapidly advancing research (Chen et al., 2020c; Caron et al., 2020; Zbontar et al., 2021), as summarized by various evaluation studies (Azizi et al., 2021; Ericsson et al., 2021), shows progress towards this goal. If successful this could displace the 'end-to-end supervised learning for each task' principle that has dominated deep learning and alleviate its data annotation cost.

However, this vision is not straightforward to achieve. In reality, different tasks often require mutually incompatible invariances (inductive biases). For example, object recognition may benefit from rotation and blur invariance; but pose-estimation and blur-estimation tasks obviously prefer rotation and blur equivariance respectively. Training a feature extractor with any given invariance will likely

harm some task of interest, as quantified recently Ericsson et al. (2022a). This has led to work on learning task-specific invariances/augmentations for self-supervision with meta-gradient (Raghu et al., 2021) or BayesOpt (Wagner et al., 2022), which is extremely expensive and cumbersome; and training feature ensembles using multiple backbones with different invariances (Xiao et al., 2021; Ericsson et al., 2022a), which is also expensive and not scalable. In this paper we therefore raise the question: *How can we learn a single general-purpose representation that efficiently supports a set of downstream tasks with conflicting, and a-priori unknown invariance requirements?*

To address these issues we explore the notion of *amortized invariance learning* in contrastive self-supervision. We parameterise the contrastive learner's neural architecture by a set of differentiable invariance hyper-parameters, such that the feature extraction process is conditioned on a particular set of invariance requirements. During contrastive pre-training, sampled augmentations correspond to observed invariance hyper-parameters. By learning this architecture on a range of augmentations, we essentially learn a low-dimensional manifold of feature extractors that is paramaterised by desired invariances. During downstream task learning, we freeze the feature extractor and learn a new readout head as well as the unknown invariance hyperparameters. Thus the invariance requirements of each downstream task are automatically detected in a way that is efficient and parameter light.

Our framework provides an interesting new approach to general purpose representation learning by supporting a range of invariances within a single feature extractor. We demonstrate this concept empirically for two different modalities of vision and audio, using SimCLR Chen et al. (2020a) and MoCo Chen et al. (2020c) as representative contrastive learners; and provide two instantiations of the amortized learning framework: a hypernetwork-based Ha et al. (2017) approach for ResNet CNNs, and a prompt learning approach for ViTs Dosovitskiy et al. (2021). We evaluate both classification and regression tasks in both many-shot and few-shot regime. Finally, we provide theoretical insights about why our amortised learning framework provides strong generalisation performance.

## 2 RELATED WORK

**Invariance Learning** Invariances have been learned by MAP (Benton et al., 2020), marginal likelihood (Immer et al., 2022), BayesOpt (Wagner et al., 2022), and meta learning (Raghu et al., 2021)-—where gradients from the validation set are backpropagated to update the invariances or augmentation choice. All these approaches are highly data and compute intensive due to the substantial effort required to train an invariance at each iteration of invariance learning. Our framework amortises the cost of invariance learning so that it is quick and easy to learn task-specific invariances downstream.

**Invariances in Self-Supervision** Self-supervised methods (Ericsson et al., 2022b) often rely on contrastive augmentations (Chen et al., 2020c). Their success has been attributed to engendering invariances (Ericsson et al., 2021; Wang & Isola, 2020; Purushwalkam & Gupta, 2020) through these augmentations, which in turn provide good inductive bias for downstream tasks. Self-supervision sometimes aspires to providing a single general purpose feature suited for all tasks in the guise of foundation models (Bommasani et al., 2021). However, studies have shown that different augmentations (invariances) are suited for different downstream tasks, with no single feature being optimal for all tasks (Ericsson et al., 2022a) and performance suffering if inappropriate invariances are provided. This leads to the tedious need to produce and combine an ensemble of features (Xiao et al., 2021; Ericsson et al., 2022a), to disentangle invariance and transformation prediction (Lee et al., 2021), or to costly task-specific self-supervised pre-training (Raghu et al., 2021; Wagner et al., 2022). Our framework breaths new life into the notion of self-supervised learning of general purpose representations by learning a parametric feature extractor that spans an easily accessible range of invariances, and provides easy support for explicit task-specific invariance estimation of downstream tasks.

**Self-Supervision in Audio and Beyond** The design of typical augmentations in computer vision benefits from a large collective body of wisdom (Chen et al., 2020b) about suitable augmentations/invariances for common tasks of interest. Besides the task-dependence (e.g., recognition vs pose-estimation) of invariance already discussed, bringing self-supervision to new domains with less prior knowledge – such as audio – often requires expensive grid search to find a good augmentation suite to use (Al-Tahan & Mohsenzadeh, 2021; Wagner et al., 2022), where each step consists of self-supervised pre-training followed by downstream task evaluation. Our framework also benefits these situations: we can simply pre-train once with a fairly unconstrained suite of augmentations, and then quickly search for those augmentations beneficial to downstream tasks in this modality.

## 3 METHODOLOGY

### 3.1 PRE-TRAINING

**Features and Invariance Descriptors** We begin by denoting a large unlabeled dataset available for pre-training by $\mathcal{D}^t = \{x_i^t\}_{i=1}^{n_t}$, where $n_t$ is the number of samples available in the raw dataset. Contrastive self-supervision typically trains a feature extractor $h(x)$ that bakes in invariance to a single pre-defined set of augmentations. We introduce the concept of an invariance descriptor $i$, that denotes whether a parameterised feature extractor $h(x; i)$ should be invariant or sensitive to $K$ possible factors of variation. This is a vector $i \in [0, 1]^K$ over $K$ possible transformations, where $i_k = 1$ and $i_k = 0$ indicate invariance and sensitivity to the $k$th factor respectively. We denote the set of binary invariance descriptors descriptors by $\mathcal{I}$, where $|\mathcal{I}| = 2^K$. [1]

**Learning an invariance-paramaterised feature extractor** Every unique invariance descriptor $i$ can be paired with a corresponding combination of stochastic augmentations, which are denoted by $\mathbb{A}_i$. To learn our invariance-paramaterised encoder $h_w(x; i)$ we extend the standard contrastive self-supervised learning paradigm. At each iteration, of contrastive learning, we sample an invariance descriptor $i$ which can thus be considered observed. We then use the corresponding augmentations to generate two views of the same example for invariance descriptor $i$, denoted by $\tilde{x}_{\mathbb{A}_i}$ and $\tilde{x}_{\mathbb{A}_i}^+$. Similarly, a set of $N^-$ negative samples denoted by $\bar{X}_{\mathbb{A}_i} = \{\bar{x}_{\mathbb{A}_i}^k\}_{k=1}^{N^-}$ are also augmented using the same augmentations, i.e. invariance descriptor $i$.

Like all contrastive learning methods, a network projection head $g_\phi(\cdot)$ projects representations to the feature space in which contrastive loss is applied. Following the convention introduced in MoCo-v2 (Chen et al., 2020c), the two views $\tilde{x}_{\mathbb{A}_i}^1$ and $\tilde{x}_{\mathbb{A}_i}^2$ are considered as input for query $q_{\mathbb{A}_i}$ and positive key $k_{\mathbb{A}_i}^+$ representations respectively. A set of encoded samples form the keys of a dictionary denoted by $\mathcal{K}_{\mathbb{A}_i} = \{k_{\mathbb{A}_i}^+, \bar{k}_{\mathbb{A}_i}^1, \bar{k}_{\mathbb{A}_i}^2 \cdots\}$. Eq. 1 shows the forward propagation pipeline of the invariance encoder backbone to generate the query and keys of $\mathcal{K}_{\mathbb{A}_i}$.

$$q_{\mathbb{A}_i} = g_\phi(h_w(\tilde{x}_{\mathbb{A}_i}; i)) \qquad k_{\mathbb{A}_i}^+ = g_\phi(h_w(\tilde{x}_{\mathbb{A}_i}^+; i)) \qquad \bar{k}_{\mathbb{A}_i}^j = g_\phi(h_w(\bar{x}_{\mathbb{A}_i}^j; i)) \qquad (1)$$

Both SimCLR and MoCo-v2, employ the contrastive InfoNCE loss Oord et al. (2018). In SimCLR variants, negative keys are from the same batch, while MoCo-based methods maintain negative keys as a queue. Finally, for a particular augmentation operation $\mathbb{A}_i$, we formulate the InfoNCE loss as:

$$\mathcal{L}_{\text{contrastive}}(q_{\mathbb{A}_i}, \mathcal{K}_{\mathbb{A}_i}) = -\log \frac{\exp\left(q_{\mathbb{A}_i} \cdot k_{\mathbb{A}_i}^+ / \tau\right)}{\sum_{j=0}^{|\mathcal{K}_{\mathbb{A}_i}|} \exp\left(q_{\mathbb{A}_i} \cdot k_{\mathbb{A}_i}^j / \tau\right)} \qquad (2)$$

where $\tau$ is a temperature hyper-parameter. In the pre-training stage, the invariance encoder and the projection head are trained using the contrastive loss governed by the contrastive method employed.

$$w^\star, \phi^\star = \arg\min_{w, \phi} \frac{1}{|\mathcal{I}|} \sum_{i_t \in \mathcal{I}} \mathcal{L}_{\text{contrastive}}(q_{\mathbb{A}_i}, \mathcal{K}_{\mathbb{A}_i}) \qquad (3)$$

In practice, we randomly sample an invariance descriptor from $\mathcal{I}$ for each batch and $w, \phi$ are learned for corresponding $\mathcal{L}_{\text{contrastive}}(q_{\mathbb{A}_i}, \mathcal{K}_{\mathbb{A}_i})$. The invariance-parameterised encoder $h^\star(x; i)$ is then transferred to a downstream task.

### 3.2 DOWNSTREAM TASK LEARNING

We next consider a set of downstream tasks $\mathcal{T}_{\text{target}}$, that may have different opposing and a priori unknown invariance requirements. We denote the training data available for a downstream task $t \in \mathcal{T}_{\text{target}}$ as $\mathcal{D}^t = \{x_i^t, y_i^t\}_{i=1}^{n_t}$. In the downstream task training stage, we employ the learned parametric encoder $h^\star(\cdot; \cdot)$ learned from the pre-training to encode data as $h^\star(x; \cdot)$. For each downstream task $t$, we follow the linear evaluation protocol by learning a prediction head $\Phi_t$, but extend it by also learning the corresponding task-wise invariance vector $i_t$. Thus the predicted output given an invariance hyper-parameter $i_t$ is $\hat{y}^t = \langle \phi_t, h^\star(x; i_t) \rangle$. For each task $t$, we find the optimal invariance

---

[1] We exclude the case where all bits correspond to 0, implying that no augmentations are applied.

hyper-parameters $i_t$ and prediction heads $\Phi_t$ by minimizing the task-specific loss of the training set,

$$i_t^\star, \Phi_t^\star = \arg\min_{i_t, \Phi_t} \frac{1}{n_t} \sum_{j=1}^{n_t} \mathcal{L}(\langle \phi_t, h^\star(x^t; i_t) \rangle, y_j^t). \tag{4}$$

**Quantised Invariance Learning** We remark that invariance parameters learned for downstream tasks are continuous vectors $i \in [0,1]^K$ in our model. In the previous pre-training phase, all observed occurrences of $i$ are discrete $i \in \{0,1\}^K$. However, during downstream learning continuous values are learned which can represent continuous degree of invariance. Nevertheless, we will show later than there are learning theoretic benefits for modeling $i$ as members of a discrete set. To exploit this, while retaining the ease of continuous optimisation for $i$ in downstream task learning, we can simply quantize to a desired number of bits $\bar{i}^* = Q(i^*; b)$ where $Q(\cdot; b)$ is the quantization operator that quantises each element of $i$ into a $b$-bit representation.

### 3.3 ARCHITECTURES

We next describe architectures $h(\cdot; i)$ capable of supporting invariance-paramaterised feature encoding for ResNet CNNs and ViT transformers.

**Hyper-ResNets:** To incorporate differentiable invariances, we parameterise the ResNet50 backbone in the form of a hypernetwork Ha et al. (2017), conditioned on an invariance descriptor. Previous work on generating ResNet-50 parameters using hypernetworks Mittal (2018) is tailored for supervised learning on small-scale dataset like CIFAR10. This architecture relies on multiple forward passes through the hypernetwork architecture to generate a single convolutional kernel, which leads to prohibitively slow pre-training with constrastive learning on large datasets like ImageNet. Thus, we develop a different hypernetwork architecture that can generate weights of a full ResNet50 with a single forward pass of the hypernetwork. This is easier to optimise and faster to train for contrastive learning. Details about the architectures are provided in the supplementary material A.1.1.

**Prompt-ViTs:** It is well known that ViTs are difficult to train, and extremely hyperparameter sensitive, especially for contrastive learning as discussed in Chen et al. (2021). While we were able to successfully learn invariance paramaterised ViTs with hypernetworks analogous to those described for ResNet above, these were even harder to train. We therefore developed an alternative approach based on prompt learning that was easier to train. Specifically, our invariance vectors are embedded by two-layer MLP network denoted by $l_{\text{prompt}}(\cdot)$ and then appended after ViT input tokens from the corresponding task. Therefore, features from an image $x$ are extracted with desired invariance $i$ as $h(x; i) = \text{ViT}([\text{CLS}, E(x), l_{\text{prompt}}(i)])$, where $E(x)$ denotes the image tokens with added position embedding. Thus invariance preferences are treated the same as image and class tokens. The invariance prompt guides the feature encoding of VIT as it is passed through all the attention and MLP layers together with the image tokens. Further details are given in the supplementary material A.1.2.

## 4 EXPERIMENTS: COMPUTER VISION TASKS

We evaluate our proposed framework on two widely-used contrastive learning methods: SimCLR and MoCo-v2 and with ResNet and VIT architectures.

### 4.1 AUGMENTATION GROUPS

Most contrastive learning studies use a suite of $K$ augmentations consisting of standard data augmentation strategies like random resized cropping, horizontal flipping, color jitter, gaussian blurring, etc. We consider two cases of treating these as $K$ independent invariances, and grouping several augmentations into a single invariance.

**Grouping Augmentations** For simple analysis and ease of comparison to prior work, we conduct experiments by grouping the augmentations into $K = 2$ groups as suggested by Ericsson et al. (2022a). The default set of augmentations have been divided into two groups called *Appearance* and *Spatial* augmentations. Spatial augmentations (crop, flip, scale, shear, rotate, transform) are those that mainly transform image spatially while Appearance based augmentations (greyscale, brightness, contrast, saturation, hue, blur, sharpness) are those that mainly act on the pixels of the image,

augmenting its appearance. Thus we amortise learning Appearance, Spatial, and default (combination of Appearance+Spatial) augmentations in a single feature extractor. During training, we assign invariance hyperparameters as 2-way binary vectors i = [1, 1], i = [1, 0] and i = [0, 1] for default, Spatial and Appearance based augmentations respectively.

**Individual Augmentations** In this condition, we perform amortised learning among the five default augmentations to learn invariance to all possible combinations of these augmentations. Every combination of augmentation is specified by a $K = 5$-way binary vector, indicating which augmentation is switched on. Since SimCLR Chen et al. (2020a) draws two augmentations out of the entire set, we exclude the invariance descriptors that indicate that less than two augmentations have been applied. Thus, *26 unique invariances are encoded into a single backbone*.

## 4.2 IMPLEMENTATION DETAILS

**Pre-training Datasets:** We perform self-supervised pre-training for ViT-B and ResNet50 on the 1.28M ImageNet training set Deng et al. (2009) and ImageNet-100 (a 100-category subset of ImageNet) following Chen et al. (2021) and Xiao et al. (2021) respectively. Both the models are pre-trained for 300 epochs with a batch size of 1024.

**Learning rates and Optimisers:** We find that the optimal learning rate and weight decay obtained by Chen et al. (2020c) work well for Hyper-ResNets and Prompt-ViTs, in both SimCLR and MoCO-v2/v3 experiments. We follow the optimization protocol in Chen et al. (2021) and use the AdamW optimiser along with learning rate warm-up for 40 epochs, followed by a cosine decay schedule.

**MLP heads:** Following Chen et al. (2020b; 2021), we use a 3-layer projection head and a 2-layer prediction head for both ViT-B and ResNet50. The hidden and output layers of the all projection and prediction MLPs of both architectures is 4096-d and 256-d respectively.

**Loss:** Following the protocol in Chen et al. (2021) for training ViT-B models under the MoCO-v3 framework, we abandon the memory queue and optimize Prompt-ViT models on the symmetrised contrastive loss (Caron et al., 2020; Grill et al., 2020). However, we observe that the symmetrised contrastive loss and discarding queues is not effective for training Hyper-ResNet50 models. Therefore, for ResNet models we stick to the MoCO-v2 framework, where a memory queue is used. This leads to fair comparison between different baselines for both the architectures. Additionally, we maintain a separate queue for each type of invariance encoded in the Hyper-ResNet50 model so that augmented keys corresponding to the same invariance are used for contrastive loss.

**Downstream Evaluation:** In our framework, evaluation on downstream tasks consists of supervised learning of the task-specific invariance hyperparameters and a linear classifier using backpropagation. We use the Adam optimiser, with a batch size of 256, and sweep learning rate and weight decay parameters for each downstream dataset based on its validation set. We apply weight decay only on the parameters of the linear classifier.

**Downstream tasks:** Our suite of downstream tasks consists of object recognition on standard benchmarks CIFAR10/100 (Krizhevsky et al., 2009), Caltech101 (Fei-Fei et al., 2004), Flowers (Nilsback & Zisserman, 2008), Pets (Parkhi et al., 2012), DTD (Cimpoi et al., 2014), CUB200 (Wah et al., 2011), as well as a set of spatially sensitive tasks including facial landmark detection on 300W SAG (2016), and CelebA Liu et al. (2015), and pose estimation on Leeds Sports Pose Johnson & Everingham (2010). More details can be found in A.2.

**Few shot downstream evaluation** We also evaluate the pre-trained networks on various few-shot learning benchmarks: FC100 (Oreshkin et al., 2018), Caltech-UCSD Birds (CUB200), and Plant Disease Mohanty et al. (2016). We also show results for few-shot regression problems on 300w, Leeds Sports Pose and CelebA datasets, where we repeatedly sampled a small subset of training data to create a low-shot training set. More details can be found in A.2.

**Competitors** We compare our Amortised representation learning framework based on Hyper-ResNet and Prompt-VIT as backbones (denoted AI-SimCLR, etc) with default SimCLR and MoCo alternatives. We also compare with SimCLR and MoCo variants that we re-trained to specialise in Appearance and Spatial group augmentations, (denoted A- and S-), and two state of the art ensemble-based alternatives LooC (Xiao et al., 2021) and AugSelf (Lee et al., 2021), with the same pretraining setting as us.

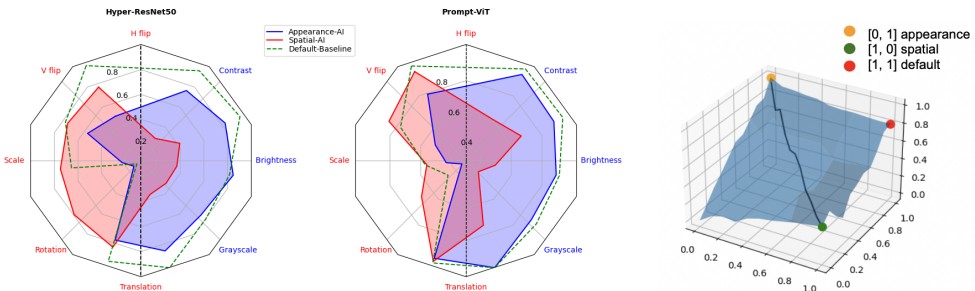

Figure 1: Left: Radar plots comparing strengths of 5 Appearance (left) and 5 Spatial invariances (right) for default ResNet50-MoCov2 and ViT-MoCov3 (green dots) vs our corresponding amortised models. By varying a *runtime* invariance parameter, a single feature extractor can provide one or other group of invariances on demand. Right: While training was performed on discrete invariance vectors (corners), we can interpolate smoothly between different invariances (vertical axis) by continuously varying the invariance parameter (horizontal axes).

### 4.2.1   RESULTS

**Can we successfully amortise invariance learning?**   We investigate this question for Appearance and Spatial invariance groups explained in Section 4.1. Specifically, we follow Ericsson et al. (2022a) in measuring the invariance between original and augmented features using the cosine similarity in a normalised feature space between input images and their augmented counterpart. A high cosine similarity indicates high invariance, and vice-versa. Using this measure, we compare the invariances learned by (1) a default MoCo model, and our single amortised model fed with (2) Appearance ($i = [1, 0]$) hyperparameter, and (3) Spatial ($i = [0, 1]$) hyperparameter. We also compare two MoCo models re-trained from scratch to specialize on the Appearance and Spatial invariance groups, following Ericsson et al. (2022a). Full details are given in Tables 6 and 7 in the appendix, and a summary of the first three comparisons in Figure 1(left). From the figure we can see that: (i) Our amortised invariance learner can access comparably strong invariances to the default model where desired (convex hull of both colored regions is similar to the dashed region). More importantly, (ii) while the default model is fixed and cannot change any (in)variances without retraining, our amortised invariance learner can *dial-down* invariances on command. For example, the Appearance model increases sensitivity to flipping. We will show that this is a useful capability for a general purpose representation when downstream tasks contain both pose estimation and classification, that require conflicting spatial invariances.

**Can We Continuously Scale Invariance Strength?**   Recall that during training, our model observed three invariance hyperparameter vectors $\{[0, 1], [1, 0], [1, 1]\}$. However, once trained we can easily interpolate along the 2d-manifold of Appearance and Spatial group invariances. Figure 1(right) illustrates this for amortised ResNet50 and grayscale invariance within the Appearance group. We can see that interpolating between Spatial $[1, 0]$ and Appearance $[0, 1]$ parameter vectors leads to a corresponding smooth increase in grayscale invariance. We expect that if a downstream task benefits from grayscale invariance it will essentially perform gradient ascent on this surface to reach the Appearance $[0, 1]$ corner. We show similar plots for Spatial amortised ResNet50 and amortised variants of ViT in the appendix. (Figure 4)

**Does amortised invariance learning benefit downstream tasks?**   We next evaluate a suite of downstream learning tasks. We compare (1) default SimCLR and MoCo models, (2) variants re-trained to specialise in Appearance and Spatial invariances (Ericsson et al., 2022a) (See C), (3) For MoCo CNNs, we compare LooC (Xiao et al., 2021) and AugSelf (Lee et al., 2021), which are ensemble based approaches to supporting multiple invariances, (4) Fine-tuning (FT) and fine-tuning with time constrained to match our method (FT-*), and (5) Our amortised framework including linear readout and invariance learning for each downstream task (denoted AI-SimCLR/MoCo etc).

From the results in Table 1, we can see that (i) our Hypernet approaches for ResNet50 have comparable or better performance compared to baselines. This is especially the case for the regression

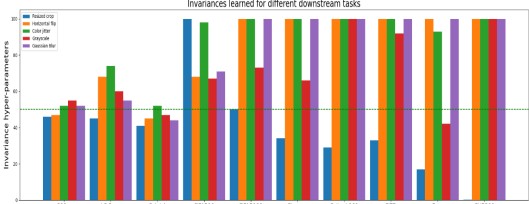 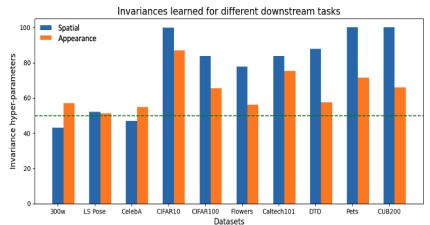

Figure 2: Learned invariance vectors for AI-SimCLR based on ResNet50 models for the suite of downstream tasks when using 2-invariance (right) and 5-invariance (left) condition.

|  | Methods | 300w | LS Pose | CelebA | CIFAR10 | CIFAR100 | Flowers | Caltech 101 | DTD | Pets | CUB | Avg. | Rank |
|---|---|---|---|---|---|---|---|---|---|---|---|---|---|
| MoCo | MoCo | 85.5 | 58.7 | 61.0 | 84.6 | 61.6 | 82.4 | 77.3 | 64.5 | 70.1 | 32.2 | 67.8 | 3.8 |
|  | MoCo - FT* | 87.7 | 61.6 | 72.1 | 85.1 | **65.1** | 80.4 | 78.0 | **69.5** | **75.8** | 40.1 | 71.5 | 2.2 |
|  | S-MoCo | 74.7 | 46.1 | 49.0 | 68.8 | 41.7 | 51.6 | 53.8 | 62.5 | 56.3 | 31.3 | 53.4 | 5.5 |
|  | A-MoCo | 78.5 | 49.2 | 62.0 | 75.0 | 37.4 | 18.8 | 43.8 | 56.3 | 39.6 | 17.6 | 47.8 | 5.5 |
|  | AugSelf+ | 77.3 | 63.9 | 77.0 | **85.3** | 63.9 | **85.7** | 78.9 | 66.2 | 73.5 | 37.0 | 70.9 | 2.5 |
|  | LooC* | – | – | – | – | – | – | – | – | – | 39.6 | – | – |
|  | AI-MoCo | **90.0** | **65.2** | **82.0** | 81.3 | 64.6 | 81.3 | 78.4 | 68.8 | 74.0 | **41.4** | **72.7** | **1.9** |
| SimCLR | SimCLR | 51.3 | 54.5 | 61.0 | 81.8 | 61.4 | 66.6 | 71.9 | 51.5 | 69.1 | 37.9 | 60.7 | 4.0 |
|  | SimCLR - FT* | 57.6 | 58.1 | 64.0 | **83.7** | **63.4** | **68.1** | **73.4** | 53.8 | 70.4 | **39.8** | 66.2 | 2.0 |
|  | S-SimCLR | 28.4 | 48.4 | 54.9 | 75.0 | 60.7 | 61.8 | 52.8 | 43.8 | 42.9 | 31.3 | 50.0 | 5.3 |
|  | A-SimCLR | 67.6 | 58.2 | 72.5 | 61.5 | 40.0 | 50.0 | 43.7 | 25.0 | 29.8 | 18.8 | 46.7 | 5.2 |
|  | AI-SimCLR (2) | 87.1 | **65.5** | 75.3 | 83.0 | 62.5 | 67.9 | 70.8 | 52.8 | 68.8 | 37.5 | 67.1 | 3.0 |
|  | AI-SimCLR (5) | **88.0** | 65.0 | **77.2** | 83.9 | 63.1 | 68.3 | 74.2 | 53.7 | 69.5 | 38.6 | 68.1 | 1.6 |

Table 1: Downstream performance of MoCo (top) and SimCLR (below) based ResNet50 models pre-trained on ImageNet-100. The first three results columns are regression tasks ($R^2$, %); the last six are classification (accuracy, %). * numbers taken from Xiao et al. (2021). + numbers for classification taken from Lee et al. (2021), rest are our runs. The invariances learned by our amortised models AI-SimCLR(2), AI-SimCLR(5) for each downstream task are indicated in Figure 2 (right and left respectively).

datasets, where the results are often dramatically better than baselines. EG: +21% for hypernet vs baseline on MoCo based CelebA, +12.7% for hypernet vs AugSelf on MoCo based 300w.

To understand how these strong results are achieved, we report the Appearance and Spatial invariances estimated by our framework in the case of SimCLR for each downstream task in Figure 2(right). The results show that while the invariance strengths learned vary continuously across tasks, there is a systematic difference: Classification tasks (right seven) prefer stronger invariances overall, and a greater Spatial- than Appearance invariance. Meanwhile regression tasks (left three) prefer more moderate invariances with similar strength or a tendency towards Appearance invariance. Invariance parameters learned for AI-MoCo models are shown in Table 10 in the appendix.

**Can we scale to learning more invariances?** We next repeat the above experiments for the case of SimCLR, using five invariances (Sec 4.1) rather than the two considered before. The results in Table 1 for AI-SimCLR(2) vs AI-SimCLR(5) show that indeed using more distinct invariances is possible, and often improves performance compared to the case of two invariances.

**Does amortised invariance learning benefit few-shot learning tasks?** To answer this question we focused on MoCo-v2 CNN models trained on ImageNet100. For classification tasks, we followed Lee et al. (2021); Xiao et al. (2021) in sampling $C$-way $K$-shot epsiodes from the target problem and training linear readouts (and invariances for the case of our method) for each episode. For regression tasks we repeatedly sampled 5%, and 20% to generate low-shot training sets. From the results in Tables 2, we can see that our AI-MoCo framework usually performs better than all competitors, with substantial margins in several cases, for example 12% and improvement on default MoCo for 5-way/5-shot flowers classification dataset, and 10% $R^2$ improvement on AugSelf in CelebA 20% low-shot regression. Invariances hyper-parameters for few-shot classification tasks and few-shot regression tasks are shown in are shown in Table 15 in the appendix. Results for few-shot learning on audio are given in the Supplementary.

**Does amortised invariance learning benefit downstream tasks when using transformer architectures?** The experiments described so far have used ResNet50 CNNs. To investigate the impact of amortised invariance learning on transformers, we apply our framework to ImageNet-1k pretrained

| Methods | CUB | | Flowers | | FC 100 | | Plant Disease | | 300w | | LS Pose | | CelebA | | Rank |
|---|---|---|---|---|---|---|---|---|---|---|---|---|---|---|---|
| | (5, 1) | (5, 5) | (5, 1) | (5, 5) | (5, 1) | (5, 5) | (5, 1) | (5, 5) | $s = 0.05$ | $s = 0.2$ | $s = 0.05$ | $s = 0.2$ | $s = 0.05$ | $s = 0.2$ | |
| MoCo | 41.0 | 56.9 | 69.6 | 76.4 | 31.7 | 43.9 | 65.7 | 85.0 | 39.0 | 50.1 | 54.2 | 60.3 | 40.2 | 52.3 | 3.2 |
| MoCo - FT | 37.8 | 52.5 | 66.5 | 73.5 | 29.4 | 40.8 | 61.0 | 80.3 | 38.2 | 45.0 | 49.0 | 56.0 | 37.2 | 48.2 | 4.3 |
| S-MoCo | 36.0 | 46.5 | 69.6 | 70.0 | 26.0 | 35.6 | 64.4 | 76.8 | 12.1 | 20.7 | 38.9 | 43.7 | 30.2 | 44.2 | 5.5 |
| A-MoCo | 24.1 | 34.7 | 24.4 | 36.4 | 21.0 | 21.2 | 21.6 | 36.6 | 36.5 | 40.8 | 43.2 | 48.2 | 43.8 | 66.3 | 5.3 |
| AugSelf | 44.2 | 57.4 | 76.0 | 85.6 | 35.0 | **48.8** | 71.8 | 87.8 | 42.0 | 51.8 | 53.8 | 60.1 | 53.2 | 66.3 | 2.1 |
| LooC | – | – | 70.9 | 80.8 | – | – | – | – | – | – | – | – | – | – | – |
| AI-MoCo | **45.0** | **58.0** | **76.7** | **88.7** | **37.4** | 48.4 | **72.6 0** | **89.1** | **49.2** | **57.9** | **55.3** | **62.0** | **56.** | **76.0** | **1.1** |

Table 2: Few-shot classification and regression accuracy (%, $R^2$) of our AI-MoCo based on ResNet50 models pretrained on ImageNet-100. Values are reported with 95% confidence intervals averaged over 2000 episodes on FC100, CUB200, and Plant Disease. (N, K) denotes N-way K-shot tasks. For regression tasks (300w, LS Pose, CelebA), we report downstream performance for different splits with train proportion given by $s$. More details are given in Tab. 11 and 12

| Methods | 300w | LS Pose | CelebA | CIFAR10 | CIFAR100 | Flowers | Caltech 101 | DTD | Pets | CUB | Avg. | Rank |
|---|---|---|---|---|---|---|---|---|---|---|---|---|
| MoCo-v3 | 81.6 | 59.1 | 78.0 | 94.8 | 63.4 | **87.7** | 83.5 | 61.1 | 79.4 | 26.5 | 71.5 | 2.7 |
| MoCo-v3 - FT* | 85.7 | 64.7 | 82.0 | **95.8** | **68.8** | 86.8 | **84.3** | **62.7** | **82.0** | 28.0 | **74.1** | 1.6 |
| S - MoCo-v3 | 50.9 | 48.1 | 72.0 | 74.3 | 62.1 | 78.9 | 72.1 | 59.3 | 73.0 | 21.4 | 61.2 | 4.6 |
| A - MoCo-v3 | 78.9 | 60.6 | 81.0 | 79.2 | 59.2 | 70.3 | 70.5 | 53.6 | 79.3 | 24.7 | 65.7 | 4.2 |
| AI-MoCo-v3 | **89.0** | **67.0** | **84.0** | 93.8 | 63.7 | 87.5 | 81.3 | 60.4 | 81.5 | **28.2** | 73.7 | 1.9 |

Table 3: Downstream performance of AI-MoCo-v3 based on ViTs pretrained on ImageNet-1k for many-shot classification (accuracy, %) and regression ($R^2$, %).

MoCo-v3 VIT-B, and repeat the many-shot and few-shot learning experiments above with this architecture. From the results in Table 3, and few-shot results in Table 5, we can see that the outcomes are broadly consistent with those of the CNNs. Our AI-VIT performs comparably or better than conventional MoCo-v3 VITs. In many cases it performs substantially better, e.g., $18\%$ and $19\%$ improvement in $R^2$ for 5% low-shot regression on 300w and LSP respectively. Invariance prompts learned for AI-ViT models are shown in Table 10 in the appendix.

**How do amortised invariances compare to Fine-Tuning?** Our framework provides a new operating point between linear readout and fine-tuning in terms of expressivity and efficiency for adapting to a downstream task. To explore this point, we compare our framework with fine-tuning variants that update 1, 2, 3 or all 4 ResNet blocks in Figure 3. The figure also shows the Pareto front of accuracy vs parameter/clock time cost. AI-MoCo with ResNet50 only requires updating essentially as few parameters as linear readout, and much less than FT; and it provides greater accuracy for less time compared to FT. In both cases AI-MoCo dominates a section of the Pareto front.

## 5 THEORETICAL ANALYSIS

We finally provide some theoretical analysis to give insight into the value and empirical efficacy of our approach when applied to novel downstream tasks not seen during pre-training. Specifically, for downstream tasks, our amortised invariance framework admits the generalisation bound below.

**Theorem 5.1.** *For 1-Lipschitz loss function, $\mathcal{L}$, taking values in $[0, M]$, if for all $\phi$ we have that $\|\phi\| \leq B$ and $\|f_\phi(x)\| \leq X$, the following holds with probability at least $1 - \delta$*

$$\mathbb{E}_{x^t, y^t}[\mathcal{L}(\hat{y}^t, y^t)] \leq \frac{1}{n_t} \sum_{j=1}^{n_t} \mathcal{L}(\hat{y}_j^t, y_j^t) + \frac{2\sqrt{2c}XB}{\sqrt{n_t}} + 3M\sqrt{\frac{\ln(|I|/\delta)}{2n_t}},$$

*where $I = \{0, 1\}^d$ is the space of possible invariance hyperparameters and $c$ is the number of classes.*

The proof is in the appendix along with corresponding existing theorems for (i) simple linear readout and (ii) fine-tuning alternatives. Comparing Theorem 5.1 with the alternatives shows that the overfitting behaviour (generalisation error) of our approach when applied to novel tasks scales similarly to conventional linear models (i.e., with $\frac{2XB}{\sqrt{n}}$). However due to the parameterised feature extractor we have a potential to obtain a much better fit on the training data (reduced first term on RHS) at a comparatively limited cost to complexity (third term on RHS). In contrast, improving train data fit by end-to-end fine-tuning of a deep network to adapt to task-specific invariance requirements

| Methods | CUB | | Flowers | | FC 100 | | Plant Disease | | 300w | | LS Pose | | CelebA | | Rank |
|---|---|---|---|---|---|---|---|---|---|---|---|---|---|---|---|
| | (5, 1) | (5, 5) | (5, 1) | (5, 5) | (5, 1) | (5, 5) | (5, 1) | (5, 5) | $s = 0.05$ | $s = 0.2$ | $s = 0.05$ | $s = 0.2$ | $s = 0.05$ | $s = 0.2$ | |
| MoCo-v3 | **65.8** | 77.0 | 83.8 | 91.6 | 65.2 | **80.8** | 79.2 | **91.4** | 15.0 | 65.3 | 29.1 | 53.3 | 53.0 | 68.7 | 2.3 |
| MoCo-v3 - FT | 62.8 | 66.7 | 81.8 | 88.8 | 61.1 | 72.4 | 72.2 | 87.9 | 11.0 | 55.0 | 24.0 | 45.0 | 50.0 | 67.0 | 1.6 |
| AI-MoCo-v3 | 65.6 | **77.2** | **84.2** | **92.2** | **67.8** | 79.8 | 76.8 | 84.6 | **33.1** | **74.6** | **48.0** | **56.0** | **55.4** | **72.3** | **1.3** |

Table 5: Few-shot classification and regression accuracy (%, $R^2$) of the AI-MoCo-v3 for ViTs pretrained on ImageNet-1k. Values are reported with 95% confidence intervals averaged over 2000 episodes on FC100, CUB200, and Plant Disease. (N, K) denotes N-way K-shot tasks. For regression tasks (300w, LS Pose, CelebA), we report downstream performance for different splits with train proportion given by $s$. Full results for regression and classification are in Tab13 and Tab 14

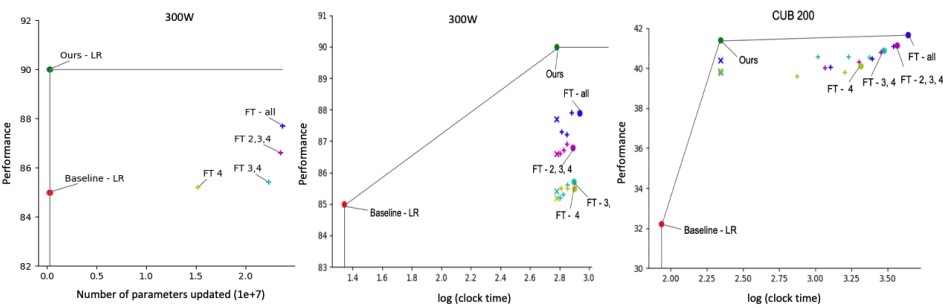

Figure 3: Comparison of linear readout (LR), AI-MoCo (Ours), and fine-tuned MoCo (FT) in terms of parameter update cost (left) and clock time cost (mid, right) vs performance. Here, we present the pareto front for two datasets: 300W (regression) and CUB 200 (classification). (x) denotes corresponding FT baseline run with time constraint while (+) denotes FT baseline for intermediate iterations. We present pareto fronts for more datasets in the appendix.

induces a third complexity term that depends exponentially on the depth of the network due to the product of norms of the weights in each layer (Bartlett et al., 2017; Golowich et al., 2018; Long & Sedghi, 2020). This analysis requires working with a discrete space of invariances, while most of our experiments have worked with continuous invariance learning for simplicity. As mentioned in Section 3.2, we can trivially discretize our estimated invariances, and we show that invariance discretization does not substantially affect our results in Table 9.

Bringing all this together, we illustrate the significance by instantiating all the terms in Theorem 5.1 to compute the guaranteed worst case generalisation error for our model and the alternatives in Table 4. The FT baseline has a vacuous bound with that only guarantees an error rate $\gg 1$. The linear model can provide an error rate guarantee, while our AI-MoCo provides a stronger guaranteed error rate thanks to the trade-off discussed above. Altogether this analysis shows that our framework provides an exciting new operating point in the bias-variance trade-off compared to the established paradigms of linear classifiers and end-to-end deep learning. See also Sec F for further details and discussion.

| Method | Guaranteed Error ($\downarrow$) |
|---|---|
| LR-MoCo | 0.78 |
| AI-MoCo | **0.67** |
| FT-MoCo | $\gg 1$ |

Table 4: Guaranteed generalistaion error for CIFAR-10

## 6 CONCLUSION

We have introduced the concept of amortised invariance learning, and shown that a manifold spanning multiple invariances (up to $K = 7$ dimensional in our experiments) can be pre-learned by a single feature extractor of either CNN or VIT types. Our amortised extractor provides an effective general purpose representation that can be transferred to support diverse downstream tasks that cannot be supported by a single conventional contrastively trained representation. With our parametric representation, each downstream task can rapidly select an appropriate invariance in an efficient and parameter-light way. This leads to strong improvements across a range of classification and regression tasks in the few-and many-shot regimes. Amortised invariances provide an exciting new direction of study for general purpose features suited for diverse downstream tasks.

## 7 ACKNOWLEDGEMENTS

Ruchika Chavhan was supported by Samsung AI Research, Cambridge.

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

# A   APPENDIX

## A.1   ADDITIONAL IMPLEMENTATION DETAILS

### A.1.1   HYPER-RESNET50

Hypernetworks Ha et al. (2017) are small neural networks that are optimised to generate parameters for another network (also called as the 'main' network'). Typically, they are designed as 2-layer MLPs conditioned on an embedding vector that describes the entire weights of a given layer. Therefore, hypernetworks are excellent choices for conditioning neural network weights and inculcate desired properties in features. In this paper, we use a hypernetwork conditioned on invariance descriptors to generate ResNet50 parameters that exhibit desired invariances in features.

**Previous Implementation for ResNet50**: In Ha et al. (2017), the network architecture is assumed to consist of kernels with the same dimensions. Taking this into consideration, this work also introduces a formulation to generate weights of deep convolutional network architecture that consists of kernels of varying dimensions. Most ResNet variants consist of kernel dimensions are typically multiples of a standard size. For example, in ResNet50, each kernel dimension is a multiple of 64. To modify hypernetworks for this architecture, Mittal (2018); Ha et al. (2017) propose to generate kernels of a basic size of $64 \times 64 \times 3 \times 3$ and concatenate multiple basic kernels together to form a larger kernel. Each basic kernel is generated by a unique $z$ embedding, and to generate larger kernels, more such embeddings are passed to the hypernetwork. For example, if we need to generate a kernel of dimension $64m \times 64n \times 3 \times 3$, $m \times n$ learnable embeddings are passed to the hypernetwork and then all these kernels are concatenated to form a kernel of the desired size. Therefore, this architecture relies on multiple forward passes through the hypernetwork to generate a single kernel, which slows down pre-training with contrastive learning on large datasets. This method requires less memory but is slower to train due to the recursive manner of weight generation. To tackle this, we adopt a block-specific hypernetwork design that can generate weights for a kernel in a single forward pass.

**Single forward-pass Hypernetwork:** Unlike Ha et al. (2017), we propose to share a embedding vector generated from invariance descriptors for each residual block in the ResNet architecture. More specifically, we learn block-specific embeddings as a function $z : \mathcal{I} \to \mathbf{R}^d$, $d$ is the dimension of the embedding vector and $\mathcal{I}$ denotes the set of invariances descriptors. We change previous implementation by abondoning a single hypernetwork to generate all kernels of the architecture. Rather, we propose to learn one hypernetwork for each kernel in a residual block given the embedding for that particular block. Thus, this architecture requires only one forward pass through a particular hypernetwork. Pytorch code has been shown in 1. Weights for each kernel are generated in a *fan-out* fashion, thus trading off between more memory and speed. However, we observe that this model is faster and easier to train.

### A.1.2   PROMPTVIT

It is well known that ViTs are difficult to train, especially for contrastive learning as demonstrated by Chen et al. (2021). Downstream performance and training stability with ViTs for contrastive learning is highly sensitive to the choice of hyper-parameters. In our initial experiments, we empirically realise that a ViT-hypernetwork architecture similar to the one designed for ResNet 50 is extremely difficult to train. Therefore, we adopt a prompt learning based approach to amortize invariance learning in ViTs, such that a range of invariances can be embedded into generated features. Specifically, binary invariance vectors are projected onto an embedding space by a two-layer invariance projector network denoted by $l_{\text{prompt}}(\cdot)$ and then appended after ViT input tokens from the corresponding task. In practice, we observe that averaging over multiple invariance projectors followed by concatenation is better for optimization and leads to a more encoding of a range of invariances.

Therefore, features with desired invariance are extracted as $\text{ViT}([\text{CLS}, E(I_{A_i})), l_{\text{prompt}}(i)])$, where $A_i$ is the set of augmentations corresponding to invariance descriptor $i$ and $E(I_{A_i})])$ denotes the image tokens with added position embedding. Code snippet in 2. In the vision transformer, invariance prompts are treated the same as image tokens and the class token as they pass all attention layers and MLP layers. The knowledge required to learn desired invariances is further utilized through the attention mechanism, where image and class tokens can extract suitable information from the invariance tokens.

| Model | Resized crop | H flip | V flip | Scale | Rotation | Translation | Grayscale | Brightness | Contrast | Saturation | Hue |
|---|---|---|---|---|---|---|---|---|---|---|---|
| A-Baseline | 0.03 | 0.63 | 0.59 | 0.26 | 0.09 | 0.71 | 1.00 | 0.88 | 0.86 | 0.98 | 0.99 |
| A-AI | 0.04 | 0.48 | 0.55 | 0.22 | 0.14 | 0.74 | 0.82 | 0.72 | 0.82 | 0.81 | 0.74 |
| S-baseline | 0.25 | 0.96 | 0.87 | 0.26 | 0.70 | 0.95 | 0.65 | 0.43 | 0.35 | 0.60 | 0.42 |
| S-AI | 0.26 | 0.75 | 0.73 | 0.72 | 0.76 | 0.80 | 0.35 | 0.34 | 0.36 | 0.41 | 0.28 |
| De-Baseline | 0.19 | 0.95 | 0.75 | 0.63 | 0.11 | 0.92 | 0.96 | 0.77 | 0.79 | 0.94 | 0.93 |
| De-AI | 0.28 | 0.81 | 0.81 | 0.62 | 0.15 | 0.76 | 0.81 | 0.73 | 0.78 | 0.75 | 0.79 |

Table 6: ImageNet-100– pre-trained AI-MoCO based on Hyper-ResNet50 (300 epochs) evaluated on invariances to transforms on ImageNet-100 validation images

| Model | Resized crop | H flip | V flip | Scale | Rotation | Translation | Grayscale | Brightness | Contrast | Saturation | Hue |
|---|---|---|---|---|---|---|---|---|---|---|---|
| A-Baseline | 0.17 | 0.73 | 0.58 | 0.45 | 0.32 | 0.89 | 0.95 | 0.81 | 0.83 | 0.91 | 0.90 |
| A-AI | 0.14 | 0.78 | 0.52 | 0.43 | 0.34 | 0.94 | 0.99 | 0.85 | 0.87 | 0.90 | 0.94 |
| S-baseline | 0.26 | 0.99 | 0.91 | 0.58 | 0.65 | 0.99 | 0.79 | 0.53 | 0.96 | 0.96 | 0.89 |
| S-AI | 0.21 | 0.94 | 0.84 | 0.55 | 0.66 | 0.96 | 0.72 | 0.41 | 0.49 | 0.68 | 0.61 |
| De-Baseline | 0.19 | 0.98 | 0.76 | 0.56 | 0.45 | 0.97 | 0.99 | 0.89 | 0.89 | 0.96 | 0.96 |
| De-AI | 0.21 | 0.97 | 0.79 | 0.55 | 0.36 | 0.97 | 0.94 | 0.63 | 0.74 | 0.91 | 0.84 |

Table 7: ImageNet-pre-trained AI-MoCo-v3 with Prompt-ViT (300 epochs) evaluated on invariances to transforms on 1000 ImageNet validation images

**Initialization trick:** We experimentally observe that the Prompt-Vit model, when initialized with the conventional Kaiming initialization is seemingly unstable in the first few epochs of pre-training. Since exploring principled techniques for weight initialization for the Prompt-ViT model is out of the scope of this study, we adopt a simple yet effective trick for faster and stable training. We initialize the Prompt-ViT model with an ImageNet1k-MoCO pretrained model obtained from Chen et al. (2021), which is trained to be invariant to the default set of augmentations. We then fine-tune the entire model along with $l_{prompt}(\cdot)$ and learnable class tokens to encode a range of multiple invariances.

## A.2 DATASETS AND EVALUATION METRICS: COMPUTER VISION TASKS

We evaluate our framework on seven object recognition and three spatially-sensitive regression tasks. For all datasets, train-test split protocol is adopted from Lee et al. (2021); Ericsson et al. (2022a).

**Classification tasks:** For linear evaluation benchmarks, we randomly choose validation samples in the training split for each dataset when the validation split is not officially provided. For CIFAR10, Flowers, CelebA, DTD, Pets, CUB200, we use the existing test sets and their train (or train+val) sets for cross-validation and training the linear classifier and invariance hyper-parameters. For Caltech101, we randomly select 30 images per class to form the training set and we test on the rest of the dataset. For CIFAR10, DTD, CIFAR100, and CUB200 we report top-1 accuracy and for Caltech101, Pets Flowers, mean per-class accuracy.

**Regression tasks:** On 300W, we use both indoor and outdoor sets and join 40% of the images in each set to form a test set. Similarly, for Leeds Sports Pose we use 40% of the images for testing. For both these datasets, the rest of the images are used for cross-validation and fitting the final model and invariance hyper-parameters. On 300W and CelebA we perform facial landmark regression and report the R2 regression metric and for Leeds Sports Pose we perform pose estimation and report R2.

**Few-shot benchmarks:** We use the meta-test split for FC100, and entire datasets for CUB200 and Plant Disease. For evaluating representations in few-shot benchmarks, we simply perform ridge regression using the representations generated using the invariance-hyperparameters and $NK$ support samples without finetuning and data augmentation in a N-way K-shot episode.

## B AMORTIZED INVARIANCES FOR AUDIO

**Setup** To complement the computer vision experiments, we also demonstrate our framework's application to invariance learning for audio representations. To evaluate this we build upon the recently released Meta Audio Heggan et al. (2022) benchmark, which provides few-shot recognition tasks across a range of audio datasets. We follow the data encoding and neural architecture of Meta Audio,

| Dataset | AI-SimCLR | | SimCLR |
| | Learned Invariance $i^\star$ | Accuracy | Accuracy |
|---|---|---|---|
| FDSKaggle18 | [74, 34, 41, 47, 41, 40, 36] | **40.99 ± 0.21** | 39.98 ± 0.20 |
| VoxCeleb1 | [57, 43, 44, 49, 49, 46, 49] | **43.01 ± 0.20** | 36.91 ± 0.19 |
| ESC-50 | [62, 49, 42, 59, 39, 39, 44] | **62.14 ± 0.20** | 60.85 ± 0.20 |
| Nsynth | [76, 24, 39, 48, 40, 45, 24] | **89.49 ± 0.14** | 88.06 ± 0.15 |

Table 8: Quantitative results for downstream few-shot audio recognition tasks from Meta Audio Heggan et al. (2022) benchmark suite, after training AI-SimCLR for ResNet18 on FSD50K.

using spectogram encodings of the audio clips and ResNet18 feature extraction on the spectograms. In this case we conduct SimCLR-based pre-training on FSD50k Fonseca et al. (2021), and then deploy the learned representation for few-shot recognition on four of the target datasets provided by Meta Audio Heggan et al. (2022), covering diverse audio types including environmental noises (ESC-50) Piczak (2015), instruments (NSynth) Engel et al. (2017), human voices (VoxCeleb1) Nagrani et al. (2017), and an eclectic mixture (FSDKaggle18) Fonseca et al. (2018). We use the $K = 7$ audio augmentation types suggested by CLAR Al-Tahan & Mohsenzadeh (2021), meaning that our feature extractor spans 120 unique invariance combinations. More details on datasets and augmentations in A.2 and C.1.

**Results** The results in Table 8 compare the downstream few-shot recognition performance of our AI-SimCLR with vanilla SimCLR. They show that we outperform conventional SimCLR, with substantial margins in some places (E.g., for VoxCeleb). We also report the learned invariance parameters (in %) for each target dataset. We can see that (i) they deviate substantially from a full vector of 1s that corresponds to SimCLR, (ii) they deviate substantially from each other, demonstrating the value of our ability to efficiently tune the representation to the invariance requirements of any given task.

## B.1 DATASETS AND EVALUATION METRICS: AUDIO

We show the performance of our framework on four target datasets covered by Meta Audio namely:

- **ESC-50** Piczak (2015) This dataset consists of 2000 labeled samples of various environmental sounds categorized into 50 classes. Classes have been grouped as animal sounds, natural soundscapes and water sounds, human (non-speech) sounds, interior/domestic sounds, exterior/urban noises with 10 classes per group. Audio clips are provided in a short 5-second format.

- **NSynth** Engel et al. (2017) NSynth is a large scale dataset that consists of 306043 musical notes, each with a unique pitch, timbre, and envelope. These samples have been divided into 1006 classes formed on the basis of the source of sample.

- **VoxCeleb1** Nagrani et al. (2017): This datasets contains over 100,000 utterances for 1,251 celebrities, extracted from videos uploaded to YouTube.

- **FSDKaggle18** Fonseca et al. (2018): FSDKaggle2018 is composed of audio content collected from Freesound Font et al. (2013). This dataset consists of 11,073 samples divided into 41 classes from sounds originating from a wide range of real-world environments like musical instruments and human/animal sounds.

For all datasets, we report 5-way 1-shot classification results in Table 8.

## C AUGMENTATION POLICIES: COMPUTER VISION TASKS

Here we provide details about the augmentation policies used to train amortized versions of SimCLR and MoCo. We use the default suite of augmentation Ericsson et al. (2022a) that previous works have also employed. Spatial augmentation is a set of spatial transformations including random resized crop and random horizonal flip. Appearance based augmentation is a set of appearance changing augmentations consisting random grayscaling, random color jitter and gaussian blurring. Finally, the default augmentations is a combination of both Spatial and Appearance augmentations. In addition

| Methods | 300w | LS Pose | CelebA | CIFAR10 | CIFAR100 | Flowers | Caltech 101 | DTD | Pets | CUB | Avg. | Rank |
|---|---|---|---|---|---|---|---|---|---|---|---|---|
| MoCo-v3 | 81.6 | 59.1 | 78.0 | 94.8 | 63.4 | **87.7** | 83.5 | 61.1 | 79.4 | 26.5 | 54.1 | 2.1 |
| AI-MoCo-v3 | **89.0** | **67.0** | **84.0** | 93.8 | 63.7 | 87.5 | 81.3 | 60.4 | 81.5 | **28.2** | **58.6** | **1.4** |
| AI-MoCo-v3 (disc.) | 85.7 | 64.0 | 81.6 | 91.5 | 62.9 | 86.8 | 80.9 | 59.3 | 79.6 | 27.1 | 56.4 | 2.5 |

| Methods | 300w | LS Pose | CelebA | CIFAR10 | CIFAR100 | Flowers | Caltech 101 | DTD | Pets | CUB | Avg. | Rank |
|---|---|---|---|---|---|---|---|---|---|---|---|---|
| MoCo | 85.5 | 58.7 | 61.0 | 84.6 | 61.6 | **82.4** | 77.3 | 64.5 | 70.1 | 32.2 | 58.9 | 2.5 |
| AI-MoCo | **90.0** | **65.2** | **82.0** | 81.3 | **64.6** | 81.3 | **78.4** | **68.8** | **74.0** | **41.4** | **65.7** | **1.2** |
| AI-MoCo (disc.) | 88.7 | 64.1 | 78.3 | 80.4 | 61.9 | 80.9 | 77.1 | 66.2 | 72.1 | 38.2 | 63.5 | 2.3 |

Table 9: Downstream performance of amortized models (Prompt-ViT and Hyper-ResNet) after discretization of learned invariances. Classification tasks report accuracy (%) and regression tasks $R^2$ (%).

| Methods | 300w | LS Pose | CelebA | CIFAR10 | CIFAR100 | Flowers | Caltech 101 | DTD | Pets | CUB |
|---|---|---|---|---|---|---|---|---|---|---|
| ResNet50 | [49, 51] | [53, 54] | [43, 62] | [95, 59] | [57, 89] | [100, 96] | [85, 88] | [73, 68] | [69, 55] | [79, 77] |
| Prompt-ViT | [67, 56] | [0, 54] | [35, 61] | [100, 100] | [55, 66] | [100, 100] | [100, 0] | [83, 0] | [100, 0] | [0, 66] |

Table 10: Invariance hyper-parameters learned by amortized MoCo models on a suite of downstream tasks. Accuracies are stated in the main paper in Table 1

to the default set, both MoCo-v2 Chen et al. (2020c)and MoCo-v3 Chen et al. (2021) also use the solarization to augment pre-training images. Thus, for fair comparison, we also choose to apply solarization as an augmentation for our MoCo experiments. It is categorized as a Appearance based augmentation.

Appearance (A-) and Spatial (S-) baselines are models pre-trained with the corresponding augmentations only. Hence, they are capable of exhibiting only one type of invariance, while default baseline model is moderately invariant to both groups of augmentations.

## C.1 Augmentation Policies: Audio

We employ the augmentation strategies suggested in Al-Tahan & Mohsenzadeh (2021) for contrastive learning with audio. Augmentations introduced in this work are categorised as either frequency or temporal transformations. The frequency transforming augmentations are Pitch Shift, Fade in, Fade out, and Noise Injection. Time masking, Time shift and Time stretching are the temporal augmentations used. We construct the 7-way invariance vector in same order.

## D Additional Results

**Discreted Invariances:** In Ericsson et al. (2022a), invariance between original and augmented features was quantified using the Mahalanobis distance or cosine similarity in a normalised feature space. Low value of Mahalanobis distance and high value of cosine similarity indicates high invariance. Similarly, we investigate the invariances learned by amortized frameworks using the measurement protocol introduced in Ericsson et al. (2022a). In Table 6 and 7, we show that the proposed amortized invariance learning frameworks indeed learn invariance to augmentations by measuring the invariances of feature vectors under input transformation.

To measure invariance to a particular set of augmentations, we pass the corresponding invariance descriptor to the model and measure invariance between resulting features. For the sake of simplicity, we only report measured cosine similarity for Appearance (A), Spatial (S) and Default (De) models. We confirm that there is broad agreement between invariances learned by amortized and baseline models, hence confirming the validity of the concept of amortising invariances. A visual comparison of invariances measured for default MoCo and amortized model for Appearance and Spatial invariances is shown in Figure 1(left).

**Continuous scaling of Invariance Strength:** Similar to Section 4.2.1, we present qualitative visualization to demonstrate smooth interpolation between polar opposite invariances. We visualize this interpolation for Spatial Hyper-ResNet 50, Appearance Prompt-ViT and Spatial Prompt-ViT in Figure 4. To evaluate this curve for Spatial- model, the Spatial hyper-parameter [1,0] is passed through the model and invariance is measured between original image and rotated image. Thus, we see that maximum invariance is achieved at the respective corners for each model. The general trend is that

| Methods | 300w | | | LS Pose | | | CelebA | | | Rank |
|---|---|---|---|---|---|---|---|---|---|---|
| | $s = 0.05$ | $s = 0.1$ | $s = 0.2$ | $s = 0.05$ | $s = 0.1$ | $s = 0.2$ | $s = 0.05$ | $s = 0.1$ | $s = 0.2$ | |
| MoCo | $39.0 \pm 0.2$ | $43.6 \pm 0.4$ | $50.1 \pm 0.2$ | $54.2 \pm 0.1$ | $56.1 \pm 0.5$ | $60.3 \pm 0.6$ | $40.2 \pm 0.7$ | $43.2 \pm 0.4$ | $52.3 \pm 0.2$ | 3.1 |
| MoCo - FT | $38.2 \pm 0.1$ | $39.2 \pm 0.5$ | $45.0 \pm 0.2$ | $49.0 \pm 0.3$ | $51.0 \pm 0.2$ | $56.0 \pm 0.1$ | $37.2 \pm 0.3$ | $42.1 \pm 0.2$ | $48.2 \pm 0.3$ | 4.1 |
| S-MoCo | $12.1 \pm 0.4$ | $14.0 \pm 0.2$ | $20.7 \pm 0.3$ | $38.9 \pm 0.7$ | $43.7 \pm 0.3$ | $47.3 \pm 0.3$ | $30.2 \pm 0.3$ | $36.2 \pm 0.2$ | $44.2 \pm 0.1$ | 6.0 |
| A-MoCo | $36.5 \pm 0.3$ | $37.4 \pm 0.2$ | $40.8 \pm 0.1$ | $43.2 \pm 0.2$ | $45.7 \pm 0.2$ | $48.2 \pm 0.1$ | $43.8 \pm 0.4$ | $51.0 \pm 0.3$ | $66.3 \pm 0.3$ | 4.3 |
| AugSelf | $42.0 \pm 0.2$ | $45.7 \pm 0.5$ | $51.8 \pm 0.2$ | $53.8 \pm 0.3$ | $58.3 \pm 0.3$ | $60.1 \pm 0.8$ | $53.2 \pm 0.1$ | $58.9 \pm 0.3$ | $66.3 \pm 0.2$ | 2.3 |
| AI-MoCo | $\mathbf{49.2 \pm 0.1}$ | $\mathbf{53.1 \pm 0.2}$ | $\mathbf{57.9 \pm 0.1}$ | $\mathbf{55.3 \pm 0.1}$ | $\mathbf{59.0 \pm 0.6}$ | $\mathbf{62.0 \pm 0.3}$ | $\mathbf{56.2 \pm 0.1}$ | $\mathbf{62.3 \pm 0.3}$ | $\mathbf{76.0 \pm 0.5}$ | 1.0 |

Table 11: Few-shot regression accuracy (%) of the ResNet50 MoCo-v2 models pretrained on ImageNet-100. We test the models over 100 train-test splits given by $s$ and report average $R^2$ score with 95% confidence intervals.

| Methods | CUB | | Flowers | | FC 100 | | Plant Disease | | Average | | Rank |
|---|---|---|---|---|---|---|---|---|---|---|---|
| | (5, 1) | (5, 5) | (5, 1) | (5, 5) | (5, 1) | (5, 5) | (5, 1) | (5, 5) | (5, 1) | (5, 5) | |
| MoCo | $41.0 \pm 0.1$ | $56.9 \pm 0.1$ | $69.6 \pm 0.5$ | $76.4 \pm 0.1$ | $31.7 \pm 0.3$ | $43.9 \pm 0.4$ | $65.7 \pm 0.5$ | $85.0 \pm 0.4$ | 52.0 | 65.6 | 3.3 |
| MoCo - FT | $37.8 \pm 0.2$ | $52.5 \pm 0.2$ | $66.5 \pm 0.3$ | $73.5 \pm 0.1$ | $29.4 \pm 0.2$ | $40.8 \pm 0.2$ | $61.0 \pm 0.1$ | $80.3 \pm 0.1$ | 48.7 | 61.7 | 4.5 |
| S-MoCo | $36.0 \pm 0.3$ | $46.5 \pm 0.4$ | $69.6 \pm 0.6$ | $70.0 \pm 0.1$ | $26.0 \pm 0.2$ | $35.6 \pm 0.5$ | $64.4 \pm 0.8$ | $76.8 \pm 0.8$ | 49.0 | 57.2 | 4.9 |
| A-MoCo | $24.1 \pm 0.2$ | $34.7 \pm 0.3$ | $24.4 \pm 0.4$ | $36.4 \pm 0.5$ | $21.0 \pm 0.1$ | $21.2 \pm 0.2$ | $21.6 \pm 0.2$ | $36.6 \pm 0.3$ | 22.7 | 32.2 | 6.3 |
| AugSelf | $44.2 \pm 0.5$ | $57.4 \pm 0.5$ | $76.0 \pm 0.4$ | $85.6 \pm 0.3$ | $35.0 \pm 0.4$ | $\mathbf{48.8 \pm 0.4}$ | $71.8 \pm 0.5$ | $87.8 \pm 0.3$ | 56.8 | 69.9 | 1.9 |
| LooC | – | – | $70.9 \pm 0.3$ | $80.8 \pm 0.2$ | – | – | – | – | – | – | – |
| AI-MoCo | $\mathbf{45.0 \pm 0.1}$ | $\mathbf{58.0 \pm 0.2}$ | $\mathbf{76.7 \pm 0.2}$ | $\mathbf{88.7 \pm 0.2}$ | $\mathbf{37.4 \pm 0.1}$ | $48.4 \pm 0.14$ | $\mathbf{72.6 \pm 0.20}$ | $\mathbf{89.1 \pm 0.2}$ | 57.9 | 71.1 | 1.1 |

Table 12: Few-shot classification accuracy (%) of the AI-MoCo (Hyper-ResNet) models pretrained on ImageNet-100. Values are reported with 95% confidence intervals averaged over 2000 episodes on FC100, CUB200, and Plant Disease. (N, K) denotes N-way K-shot tasks.

| Methods | CUB | | Flowers | | FC 100 | | Plant Disease | |
|---|---|---|---|---|---|---|---|---|
| | (5, 1) | (5, 5) | (5, 1) | (5, 5) | (5, 1) | (5, 5) | (5, 1) | (5, 5) |
| MoCo-v3 | $\mathbf{65.8 \pm 0.5}$ | $77.0 \pm 0.5$ | $83.8 \pm 0.5$ | $91.6 \pm 0.3$ | $65.2 \pm 0.5$ | $\mathbf{80.8 \pm 0.5}$ | $\mathbf{79.2 \pm 0.5}$ | $\mathbf{91.4 \pm 0.4}$ |
| MoCo-v3 - FT | $62.8 \pm 0.1$ | $66.7 \pm 0.3$ | $81.8 \pm 0.2$ | $88.8 \pm 0.2$ | $61.1 \pm 0.2$ | $72.4 \pm 0.1$ | $72.2 \pm 0.3$ | $87.9 \pm 0.2$ |
| AI-MoCo-v3 | $65.6 \pm 0.7$ | $\mathbf{77.2 \pm 0.4}$ | $\mathbf{84.2 \pm 0.2}$ | $\mathbf{92.2 \pm 0.2}$ | $\mathbf{67.8 \pm 0.2}$ | $79.8 \pm 0.5$ | $76.8 \pm 0.6$ | $84.6 \pm 0.5$ |

Table 13: Few-shot classification accuracy (%) of the AI-MoCo-v3 models pretrained on ImageNet-1k. Values are reported with 95% confidence intervals averaged over 2000 episodes on FC100, CUB200, and Plant Disease. (N, K) denotes N-way K-shot tasks.

| Methods | 300w | | | LS Pose | | | CelebA | | | Rank |
|---|---|---|---|---|---|---|---|---|---|---|
| | $s = 0.05$ | $s = 0.1$ | $s = 0.2$ | $s = 0.05$ | $s = 0.1$ | $s = 0.2$ | $s = 0.05$ | $s = 0.1$ | $s = 0.2$ | |
| MoCo-v3 | $15.0 \pm 0.2$ | $33.1 \pm 0.4$ | $65.3 \pm 0.1$ | $29.1 \pm 0.4$ | $39.2 \pm 0.2$ | $53.3 \pm 0.4$ | $53.0 \pm 0.1$ | $61.2 \pm 0.5$ | $68.7 \pm 0.5$ | 3.0 |
| MoCo-v3 - FT | $11.0 \pm 0.1$ | $25.0 \pm 0.1$ | $55.0 \pm 0.2$ | $24.0 \pm 0.3$ | $34.0 + 0.2$ | $45.0 \pm 0.6$ | $50.0 + 0.6$ | $57.9 \pm 0.4$ | $67.0 \pm 0.1$ | 2.0 |
| AI-MoCo-v3 | $\mathbf{33.1 \pm 0.5}$ | $\mathbf{45.2 \pm 0.1}$ | $\mathbf{74.6 \pm 0.2}$ | $\mathbf{48.0 \pm 0.2}$ | $\mathbf{52.4 \pm 0.2}$ | $\mathbf{56.0 \pm 0.3}$ | $\mathbf{55.4 \pm 0.1}$ | $\mathbf{64.3 \pm 0.3}$ | $\mathbf{72.3 \pm 0.5}$ | 1.0 |

Table 14: Few-shot regression accuracy (%) of the AI-MoCo-v3 models pretrained on ImageNet-1k. We test the models over 100 train-test splits given by $s$ and report average $R^2$ score with 95% confidence intervals.

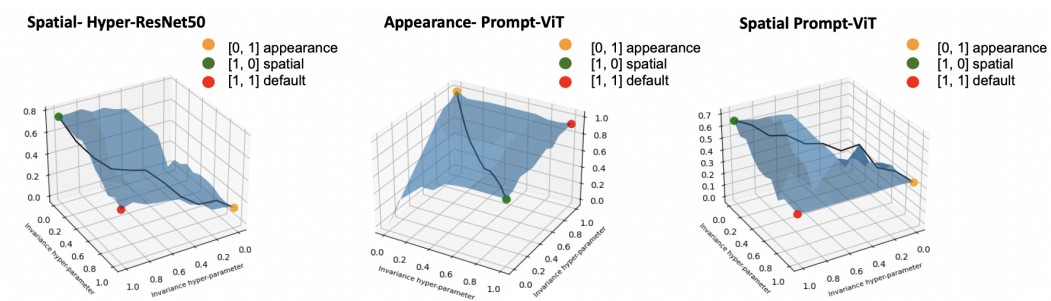

Figure 4: Qualitative visualisation of the smooth innterpolation between different invariances (vertical axis) by continuously varying the invariance parameter (horizontal axes) for Spatial Hyper-ResNet50, Appearance Prompt-ViT and Spatial Prompt-ViT.

| Methods | CUB | | Flowers | | FC 100 | | Plant Disease | |
|---|---|---|---|---|---|---|---|---|
| | (5, 1) | (5, 5) | (5, 1) | (5, 5) | (5, 1) | (5, 5) | (5, 1) | (5, 5) |
| ResNet50 | [47, 51] | [35, 51] | [77, 75] | [95, 1] | [50, 49] | [50, 49] | [50, 49] | [50, 49] |
| Prompt-ViT | [50, 51] | [50, 49] | [50, 48] | [50, 47] | [50, 49] | [50, 49] | [51, 51] | [51, 51] |

| Methods | 300w | | | LS Pose | | | CelebA | | |
|---|---|---|---|---|---|---|---|---|---|
| | $s = 0.05$ | $s = 0.1$ | $s = 0.2$ | $s = 0.05$ | $s = 0.1$ | $s = 0.2$ | $s = 0.05$ | $s = 0.1$ | $s = 0.2$ |
| ResNet50 | [47, 51] | [43, 52] | [43, 56] | [51, 53] | [50, 56] | [49, 60] | [48, 54] | [46, 56] | [42, 60] |
| Prompt-ViT | [49, 51] | [48, 51] | [58, 52] | [45, 52] | [40, 51] | [50, 53] | [52, 57] | [54, 53] | [46, 57] |

Table 15: Invariance hyper-parameters learned by amortized MoCo models on few shot classification and regression tasks.

A- or S- models display high invariance for corresponding input augmentation while default models shows a moderate invariance.

**Learned Invariance Hyper-parameters:** During downstream evaluation, we initialise the invariance hyper-parameters with a vector of $0.5^K$. This implies that we provide moderate invariance to each type of invariance and let the model automatically descend to the invariances that are favored by the task. Invariance Hyper-parameters learned for MoCo models for all downstream tasks are presented in Table 10. As expected, regression based tasks tend to slightly prefer Appearance augmentations while object recognition tasks prefer the set of default or Spatial augmentations. We notice that fine-grained classifcation tasks like CUB200 prefer Appearance augmentations.

Next, we present invariance hyper-parameters learned for few-shot regression and classification tasks in Table 15. Invariances are learned episode-wise and we present the invariance parameters averaged over all episodes. For few shot experiments it can be observed that classification tasks prefer slight Spatial or default invariance. For regression tasks, Appearance invariance is prefered in most cases.

# E    COST ANALYSIS

**Pareto Front Analysis:** We previously argued that our proposed framework is essentially new operating point between linear readout and fine-tuning. In this section, we present more Pareto fronts in Figure 5 and 6 for parameter cost and clock time cost respectively. The time cost is the most visible cost to the downstream user of a feature, and the parameter cost is important because of its connection to overfitting as we will see in the Section F.

For this analysis, we consider 2 regression datasets (300W, Leeds Sports Pose) and 3 classification datasets (DTD, CIFAR10, and CUB200). In both Figure 5 and 6, the bottom right figure corresponds to the average over all the datasets. In Figure 5, it is evident that AI-MoCo dominates the section of the Pareto front for all datasets. With respect to clock time cost in Figure 6, AI-MoCo dominates for both regression datasets and CUB 200. Even though our model does not fall on the Pareto front for CIFAR10 and DTD, AI-MoCo dominates a section of the Pareto front on average.

**Downstream Memory requirements:** We provide a comparison of memory requirement for downstream training in Table 17 for both ResNet50 and ViTs. Our memory requirement is higher than baselines for ResNet50 and significantly lower for Prompt-ViTs. This is due to the fact that Hyper-ResNet50 consists of more parameters than baseline ResNet50 whereas the Prompt-ViT is comparable to baseline ViTs in terms of both architecture and number of parameters.

**Pretraining:** Pre-training is a one-off cost that is often expensive in self-supervision, but is expected to be amortised across multiple downstream problems. Our method goes further by pre-training a model that represents $k$ multiple invariances, indeed multiple *combinations* of invariances, to make them available for downstream models. We discuss the computational efficiency of pretraining our method and compare it with training one baseline model, and $2^k$ baseline models in Table 16. In this case, we provide a comparison with $k = 5$. Table 16 shows the time taken in GPU days on Tesla V100-SXM2-32GB for 300 pre-training epochs for ResNet50 and ViTs with ImageNet-100 and ImageNet-1k respectively. The results show that pre-training our model takes more time and memory than a standard MoCo model, but less than the baseline of pre-training $2^5$ unique MoCos for each combination of invariances.

| Method | Ours | | Baseline | | | |
|---|---|---|---|---|---|---|
| | RN50 | ViT | 1 RN50 | $2^5$ RN50s | 1 ViT | $2^5$ ViTs |
| Time (GPU days) | 5.07 | 29.3 | 2.9 | 92.8 | 27.3 | 874.7 |
| Memory (Gb) | 446.4 | 481.6 | 134.4 | 134.4 | 476.8 | 476.8 |

Table 16: Comparison of pretraining cost for amortized MoCo models and baselines. We assume that the models are run serially where one baseline model will require the same memory as $2^k$ baseline model. If we consider parallel training, then memory requirements for $2^k$ baselines increases drastically while clock time remains the same.

| Memory | Ours | Baseline - LR | FT - all | FT - 25% | FT - 50% | FT - 75% |
|---|---|---|---|---|---|---|
| ResNet50 | 13690 | 1452 | 10930 | 7250 | 5238 | 4420 |
| ViT | 14156 | 1452 | 18916 | 17608 | 15976 | 14468 |

Table 17: Memory requirements (in Mb) for downstream training. FT - $f$ denotes the variants of finetuning where $f\%$ of the number of layers/blocks is finetuned. In the case of ResNet50, FT - 50% indicates that the out of 4 layers, the final two layers (3 and 4) are finetuned. For ViTs, FT - 50% is the baseline where block 6 to block 12 of the architecture is finetuned. FT - all denotes the finetuning baseline where the entire model is finetuned.

## F   THEORETICAL ANALYSIS

**Generalisation Bound for Amortised Invariance Framework** We first present the proof for Theorem 5.1, which gave the generalisation bound of our amortised invariance framework.

*Proof.* There is a one-to-one mapping between elements of the finite set of invariance hyperparameters and potential feature extractors, implying that there is also a finite number, $|I|$, of potential feature extractors for a novel task. For a linear model coupled with a pre-selected feature extractor, one can apply the bound on empirical Rademacher complexity for multi-class linear models developed by Maurer (2016). Combining this with the standard Rademacher complexity-based generalisation bound (Bartlett & Mendelson) tell us that with probability at least $1 - \delta'$

$$\mathbb{E}_{x^t,y^t}[\mathcal{L}(\hat{y}^t, y^t)] \leq \frac{1}{n_t} \sum_{j=1}^{n_t} \mathcal{L}(\hat{y}_j^t, y_j^t) + \frac{2\sqrt{2c}XB}{\sqrt{n_t}} + 3M\sqrt{\frac{\ln(2/\delta')}{2n_t}}. \tag{5}$$

If one were to evaluate the above bound using every possible choice of feature extractor in $I$, then the bound can only be guaranteed to hold with probably at least $1 - |I|\delta'$, due to the union bound. To compensate, on can instead instantiate the bound with $\delta' = \delta/|I|$. Substituting this definition for $\delta'$ yields the result. □

**Generalisation Bound for Standard Linear Models** In comparison to our amortised invariance framework, the generalisation bound for a standard linear readout is given by Eq. 5. Note the difference to Theorem 5.1 is the reduced third complexity term with no dependence on $I$, as well as the fact that the two models will have different empirical risk (first loss term), which will be less in the case of our model.

**Generalisation Bound for Fine-Tuning** It is also interesting to consider one of the state of the art bounds for fine-tuning. In practice, there are no bounds for deep networks that take into account both fine-tuning and residual connections. We therefore concern ourselves with the main result of Gouk et al. (2021), given in the theorem below, which provides a result for a class of deep networks without residual connections,

$$\mathcal{F} = \{(\vec{x}, y) \mapsto \mathcal{L}(y, f(\vec{x})) : \|W_j\|_\infty \leq B_j, \|W_j^0\|_\infty \leq B_j, \|W_j - W_j^0\|_\infty \leq D_j\}.$$

**Theorem F.1.** *For all $\delta \in (0, 1)$, the expected risk of all models in $\mathcal{F}$ is, with probability $1 - \delta$, bounded by*

$$\mathbb{E}_{(\vec{x},y)}[\mathcal{L}(f(\vec{x}), y)] \leq \frac{1}{m} \sum_{i=1}^m \mathcal{L}(f(\vec{x}_i), y_i) + \frac{4\sqrt{\log(2d)}cX \sum_{j=1}^L \frac{D_j}{B_j} \prod_{j=1}^L 2B_j}{\sqrt{m}} + 3M\sqrt{\frac{\log(2/\delta)}{2m}},$$

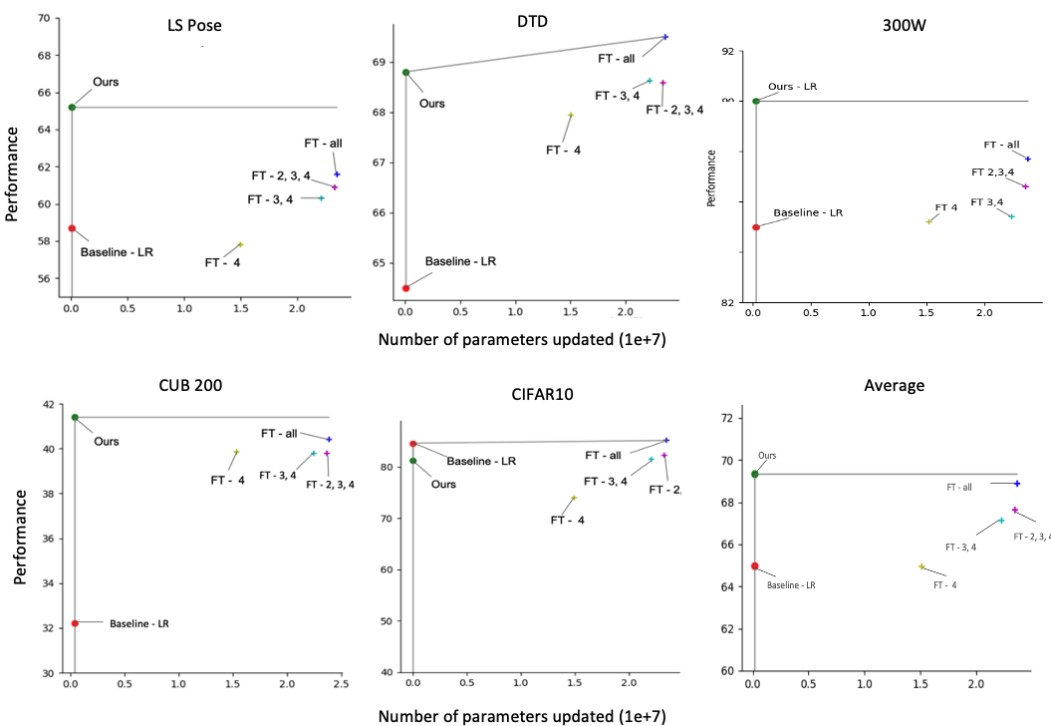

Figure 5: Comparison of linear readout (LR), AI-MoCo (Ours), and fine-tuned MoCo (FT) in terms of parameter update cost vs performance. Here, we present the pareto front for five datasets.

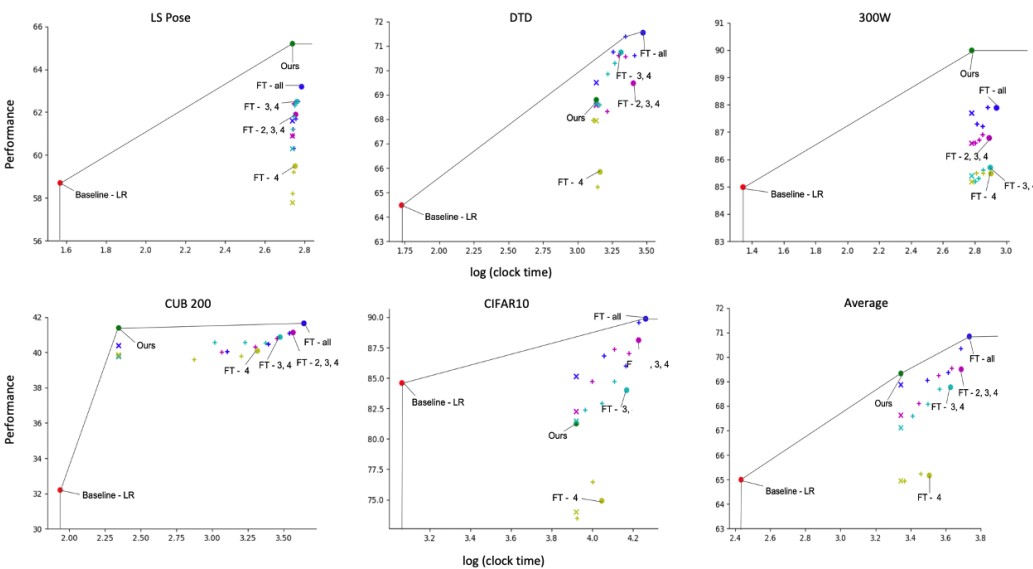

Figure 6: Comparison of linear readout (LR), AI-MoCo (Ours), and fine-tuned MoCo (FT) in terms of clock time cost vs performance. Here, we present the pareto front for five datasets. (x) denotes corresponding FT baseline run with time constraint while (+) denotes FT baseline for intermediate iterations.

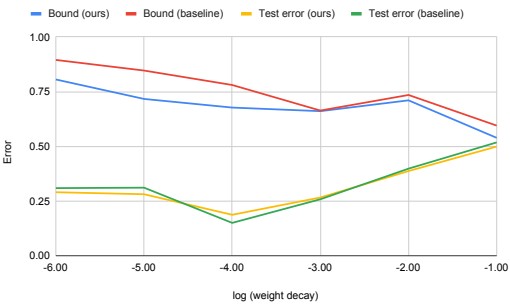

Figure 7: Generalization bound and empirical test error for a range regularisation strengths. Here, the X-axis corresponds to the log of the weight decay parameter.

*where $m$ is the number of training examples, $c$ is the number of classes, $\vec{x}_i \in \mathbb{R}^d$, $\|\vec{x}_i\|_\infty \leq X$, and it is assumed the loss function is 1-Lipschitz and takes values in $[0, M]$.*

This bound will obviously be vacuous ($\gg 1$) for any reasonable neural network, due to the exponential dependence on the depth of the network. In contrast, we demonstrate that our bound is non-vacuous in a common experimental setting based on CIFAR-10.

**Detailed Bound Instantiation Results** To expand upon the results briefly reported in Table 4, we present Figure 7, which plots the guaranteed test error and empirical test error for a range of regularisers. We can see that our model always has a tighter bound and sometimes has better empirical error.

## G CODE SNIPPETS

**Code for generating parameters of a `BottleNeck` block of the ResNet50 architecture with our modified design**

```
1
2  # Modified downsample layer
3  class DownSampleConv(nn.Module):
4      def __init__(self, stride, padding, first_dilation):
5          super().__init__()
6          self.stride = stride
7          self.padding = padding
8          self.first_dilation = first_dilation
9
10     def forward(self, x, w_down):
11         return F.conv2d(x, w_down, stride = self.stride,
12             padding = self.padding, dilation = self.first_dilation)
13
14 # Modified Bottleneck layer for Resnet50
15 class Bottleneck(nn.Module):
16     expansion = 4
17     def __init__(
18         self, inplanes, planes, stride=1, downsample=None,
19         cardinality=1, base_width=64, reduce_first=1, dilation=1,
20         first_dilation=None, act_layer=nn.ReLU,
           norm_layer=nn.BatchNorm2d, attn_layer=None,
21         aa_layer=None, drop_block=None, drop_path=None):
22
23         super(Bottleneck, self).__init__()
24
25         width = int(math.floor(planes * (base_width / 64)) * cardinality)
26         first_planes = width // reduce_first
27         outplanes = planes * self.expansion
28         first_dilation = first_dilation or dilation
```

```python
            use_aa = aa_layer is not None and (stride == 2 or first_dilation
                            != dilation)
        self.use_aa = use_aa
        self.stride = stride
        self.first_dilation = first_dilation
        self.cardinality = cardinality

        self.emb = nn.Linear(2, 16, bias=False)

        # Hypernet for conv1
        self.hypernet1 = HyperNetwork(kernel_size = 1,
                    in_size = inplanes,out_size = first_planes)

        # Hypernet for conv2
        self.hypernet2 = HyperNetwork(kernel_size = 3,
                    in_size = first_planes,out_size = width)

        # Hypernet for conv3
        self.hypernet3 = HyperNetwork(kernel_size = 1,
                    in_size = width,out_size = outplanes)

        # Hypernet for downsampling
        if downsample is not None:
            self.hypernet_down = HyperNetwork(kernel_size = 1,
                        in_size = inplanes, out_size = outplanes)

        self.bn1 = norm_layer(first_planes)
        self.act1 = act_layer(inplace=True)
        self.bn2 = norm_layer(width)
        self.drop_block = drop_block() if drop_block is not None
                else nn.Identity()
        self.act2 = act_layer(inplace=True)
        self.aa = create_aa(aa_layer, channels=width,
                stride=stride, enable=use_aa)

        self.bn3 = norm_layer(outplanes)
        self.bn_down = norm_layer(outplanes)

        self.se = create_attn(attn_layer, outplanes)

        self.act3 = act_layer(inplace=True)
        self.downsample = downsample
        self.stride = stride
        self.dilation = dilation
        self.drop_path = drop_path

    # Forward function with invariance-hyperparameters as input
    def forward(self, x, inv):
        shortcut = x

        # Block-specific embedding
        emb_ = self.emb1(inv)

        # One forward pass to generate first kernel
        w1 = self.hypernet1(emb_)
        x = F.conv2d(x, w1)
        x = self.bn1(x)
        x = self.act1(x)

        # One forward pass to generate first kernel
        w2 = self.hypernet2(emb_)
        x = F.conv2d(x, w2, stride = 1 if self.use_aa else self.stride,
            padding=self.first_dilation,
            dilation=self.first_dilation,
            groups=self.cardinality)
```

```
 93
 94            x = self.bn2(x)
 95            x = self.drop_block(x)
 96            x = self.act2(x)
 97            x = self.aa(x)
 98
 99            # One forward pass to generate first kernel
100            w3 = self.hypernet3(emb_)
101            x = F.conv2d(x, w3)
102            x = self.bn3(x)
103
104            if self.se is not None:
105                x = self.se(x)
106
107            if self.drop_path is not None:
108                x = self.drop_path(x)
109
110            # downsample [0] is convolution and downsample[1] is norm
111            if self.downsample is not None:
112                w_down = self.hypernet_down(emb_)
113                shortcut = self.downsample[1](self.downsample[0](shortcut,
                              w_down))
114            x += shortcut
115            x = self.act3(x)
116
117            return x
```

Listing 1: Code for ResNet50 `Bottleneck` modified to incorporate hypernetworks. Basic code block has been adapted from the Pytorch Image Models Library Wightman (2019)

**Code for concatenation of invariance tokens in Prompt-ViT**

```
 1
 2 # Concatenation of class and invariance tokens with invariance projectors
 3
 4 embed_dim = 768 # For ViT-Base
 5 self.inv_fc = nn.ModuleList([
 6     nn.Sequential(nn.Linear(2, self.embed_dim), nn.ReLU(),
 7     nn.Linear(self.embed_dim, self.embed_dim)) for i in range(0, 256)])
 8
 9 # We only show concatenation and position embedding of the
       vision_transformer class from MoCo-v3 https://github.com/
       facebookresearch/moco-v3/blob/main/vits.py
10
11 def _pos_embed(self, x, inv):
12     # Calculate average over multiple invariance tokens
13     inv_tokens = torch.cat([self.inv_fc[i](inv).view(1, 1, embed_dim)
14                 for i in range(0, 256)], dim = 1)
15     inv_tokens = inv_tokens.mean(1).unsqueeze(1)
16     inv_tokens = inv_tokens.expand(x.shape[0], -1, -1)
17     if self.cls_token is not None:
18         x_cls = torch.cat((self.cls_token.expand(x.shape[0], -1, -1), x),
                     dim=1)
19     x = x_cls + self.pos_embed
20     x = torch.cat([x, inv_tokens], dim = 1)
21     # Dropout
22     return self.pos_drop(x)
```

Listing 2: Code snippet for invariance token based prompting for ViTs. Basic code block has been adapted from the Pytorch Image Models Library Wightman (2019)

