# OpenReview forum: "Amortised Invariance Learning for Contrastive Self-Supervision"
_ICLR.cc/2023/Conference — ICLR 2023 poster_

### Official Review · Reviewer_SKik · 2022-10-23

**Confidence:** 4
**Correctness:** 4
**Technical Novelty And Significance:** 2
**Empirical Novelty And Significance:** 2
**Recommendation:** 6

**Clarity, Quality, Novelty And Reproducibility:**

Clarity
=====
The narrative is clear, easy to follow and enjoyable to read. The paper motivates the problem statement well, clearly explains the high-level approach and provides empirical justification. However, as mentioned above, more clarity on the downstream adaptation algorithm in the main manuscript will be beneficial for the readers.

Quality
=====
In my opinion, this works meets the bar for quality for a top-tier ML conference. The presentation is crisp, amount of contribution is non-trivial and experimental results are strong.

Novelty
======
The idea of using hyper-networks to learn the proper augmentation strategy such that right kind of invariances are preserved for a given downstream task adaptation is novel. However, I'd like the authors to clarify my comments regarding how this is different from other algorithms that also learn the optimal data augmentations for a given task before I finalize my thoughts on novelty.

Reproducibility
============
The authors provide most of the details and implementation notes that will be required to reimplement the algorithms. However, as I pointed out above, additional details on the end-to-end optimization process with hyper-networks and ViT + prompt-tuning will be helpful.


**Details Of Ethics Concerns:**

N/A.

**Strength And Weaknesses:**

Strengths
========

Narrative
-----------
The paper is written in a clear way - starting from the problem formulation and why it is important, followed by clearly motivating their proposed approach and positioning it against the baselines and other relevant methods from the literature and finally showing experimental results with different architectures and task modalities.

Practicality
-------------
Apart from only considering the formulation or theoretical properties, authors also paid attention to make sure their proposed algorithm can be trained without a lot of tweaking and training can scale to standard SSL benchmarks like ImageNet-1K. Towards this, authors modified the hyper-network formulation for ResNets. For ViTs, they proposed a different prompt tuning based formulation as they found hyper-networks + ViTs are harder to train. There are also additional tricks mentioned in the paper for ViTs which are useful for reproduction. I appreciate the effort to make sure the proposed algorithm can be used by practitioners.

Coverage
------------
Authors perform experiments using multiple classification and regression tasks for visual recognition, few-shot classification and also performed experiments in the audio domain. As mentioned before, they experimented with two different family of architectures - ViT and CNN. I believe the empirical setup is strong and helps to properly evaluate the contributions of this work.

Weakness
================
Additional details on test-time adaptation will be helpful
-----------------------------------------------------------------
More details on the test-time adaptation procedure will be helpful as that is the core contribution of this work. Along with that, some explanation of the basic hyper-network formulation will be useful as well. I have gone through the code provided in the appendix (for BottleNeck layer of ResNet) but I am unclear as to how this is plugged into the end-to-end optimization procedure. For example, I am not sure what is the shape of the invariance hyperparameters for ResNet for every block and how are those parameters getting updated.

Connecting to other types of work
-----------------------------------------
Although there has not been any work in terms of learning proper augmentations or learning a differentiable masking in the particular context of SSL, there has been prior works independently. For example, [this paper](https://arxiv.org/abs/2104.04282) learns the optimal data augmentation policy by differentiable search. [AutoAugment](https://arxiv.org/abs/1805.09501) does a similar thing using reinforcement learning. How does the proposed hyper-networks based method compare to such methods to find the right data augmentation for the given task? Can these algorithms be used to search for the augmentation strategy instead which will be in sync with the optimal invariances to learn each task?

Additional Questions
=================

* While trying to learn the right augmentation strategy for a given task, I do not see a justification why a hyper-network based strategy was used as opposed to using other ideas from the literature like straight-through estimator or using a standard meta-learning algorithm.
* While adapting to the downstream task, in Eq. 4, what will happen if we rather minimize the loss on a validation set like meta-learning? Why did you choose to minimize it on a train set instead of validation?
* Although there has been considerable effort made to reduce the computation time, still, adding hyper-networks require making three additional forward pass calls for each block in a residual network. How much additional time does these calls cause? Assuming they cause 3x, that is equivalent to performing downstream adaptation with three different augmentation strategies as hyperparameters. In that case, there is no practical advantage of this algorithm as it also tries to choose between three combinations (dorsal, ventral and both).
* What is the time difference between ResNet + hyper-network and ViT with prompt tuning? Also, my understanding is that these additional calls for hyper-networks are only needed during downstream adaptation and not pre-training - is that correct?
* When you mention ViTs are harder to train with the hyper-networks formulation, does it mean it needs more hyperparameters tuning or is the training itself unstable and leads to NaN and other problems?


**Summary Of The Paper:**

In this work, authors propose a method to dynamically learn/adjust what invariances a self-supervised representation should be robust to such that it can cater to various types of downstream tasks which may have conflicting invariances. Towards this end, they propose a hyper-network based learning method which can learn the proper invariances for a given target task. Experiments show that learning task-specific invariances (via various stochastic augmentation policies) works better compared to the original SSL models (SimCLR and MoCo) which uses a fixed augmentation policy for all tasks. The proposed method also outperforms existing works from the literature which tries to solve the same problem (e.g. LooC). Authors  perform experiments with two family of architectures - ResNets and ViT and also two task modalities - vision and audio and show improvements on most, if not all cases.

**Summary Of The Review:**

Overall, I think the paper takes a new approach to solve a well-known and important problem for self-supervised representation learning. The proposed method intuitively makes sense at a high-level and shows promise in empirical evaluations. However, I'd like for more clarifications regarding the details of the algorithm, the computational cost as well as some explanations regarding its difference with respect to a few other works from the literature. At this point, I am leaning towards marginally below acceptance threshold but I will be willing to increase my score if authors provide satisfactory responses to my comments above.

Updates after Rebuttal
---------------------------
Updated score by one point based on all the discussions between the authors and myself.

---

> ### Author Response · Authors · 2022-11-19
> **Response to Reviewer SKik**
>
> Thanks for your time and feedback!
>
> **Q1: More details on test-time adaptation.**
>
> The invariance hyper-parameter is a k-dimensional vector and is plugged into each of these hypernetworks during pre-training. For downstream task learning, we initialise the invariance vector by [0.5]^k. The invariance hyperparameter is updated using the gradients accumulated from all the hypernetworks. Essentially, we perform both model and invariance learning jointly for downstream tasks.
>
> **Q2: Connection to other other work**
> Thanks for raising this great question! AutoAugment and similar methods search for augmentation policies that enable best end-to-end learning of a supervised objective for a downstream task. In the context of our work, this can be understood as an example of a non-amortised invariance learning. Non-amortised means that the full cost of the search is would need to be paid repeatedly for each downstream task. In contrast, for our amortised framework, all the augmentations are learned once during the self-supervised pre-training step. During the actual downstream task adaptation, we can efficiently select an ideal invariance configuration from the manifold of options built into our pre-trained model by augmentations during the pre-training step. For example AutoAugment takes 5000 GPU-hours to train a new augmentation policy (~invariance) for CIFAR-10. Our amortised framework can find a good invariance for CIFAR in around 1 GPU-minute (see also the new Fig 3).
>
> **Q3:  Why hypernetwork rather than straight-through or meta-learning**
> The hypernetwork is used because of the goal of amortising the learning so that the whole manifold of invariances can be baked into a pre-trained model and made available for efficient selection by any new downstream task. In order to use meta-learning or straight through (as for AutoAugment above), it would necessitate knowing the downstream task in advance (our goal is to define a pre-trained feature suitable for many downstream tasks), and would require non-amortised learning where each downstream task has to learn the invariances from scratch at great expense. Other methods such as the cited (Raghu, NeurIPS’21) have tried to do exactly what the reviewer suggests with meta-learning and it’s super expensive due to lack of amortisation.
>
> **Q4: What about optimising downstream task invariance on the val set.**
>
> This is a great comment! We initially also felt (and still feel) that this is the best thing to do. However we tried it, and in practice it didn’t make much difference yet, so we stuck with the simpler option of just using the target task train set, rather than requiring some kind of bi-level optimisation for the target task. We think the reason it didn’t help is because the current classifier/decoder is a single linear layer that can’t seriously overfit to the target train set. If it was a deep classifier that could overfit, then it might be more helpful to train the invariance on the downstream val set.
>
> **Q5: On Computation Cost.**
>
> Please note that during adaptation we do not require to make a particular number of extra forward passes related to the number of augmentations that our hypernetwork has been trained with. During adaptation, we maintain one estimate of the invariance parameters and update it by gradient. So any “extra calls” just depend on how many gradient steps are allowed, or required to converge. The cost of this bears no particular relation to the number of invariances supported. The new Fig 3 shows examples of the cost of our method compared to linear readout and fine-tuning, and it has a favorable cost-accuracy tradeoff.
>
> We do emphasise, however, that our hypernetwork does more than simply aggregating the capabilities of k possible base networks as suggested by the reviewer. Actually, it aggregates at least 2^k base networks covering all possible unique combinations of invariances. In fact it aggregates even more than this, because our network can model continuous degrees of invariance in the whole [0,1]^k space of possible invariances.  This is illustrated in Fig 2 which shows real valued invariance vectors being estimated for different downstream tasks within the [0,1]^k space, and in Fig 3 which shows how the invariances can interpolate between corners of this space. Thus the network does substantially more than simply aggregate a handful base models as suggested.
>
> **Q6: What is the time difference between ResNet+HyperNet and VIT with prompt tuning. When are the extra calls?**: The elapsed time to solve, e.g., CIFAR-10 downstream are 8312.27 seconds and 9974 seconds  respectively.  There are no extra (number of invariance dependent) calls in the way that the reviewer suggests in either pre-training or downstream adaptation.
>
> **Q7: VIT HyperNet hard to train**: It seemed to require more hyperparameter tuning to avoid unstable training leading to NaNs, etc.

---

> > ### Comment · Reviewer_SKik · 2022-11-30
> > **Thanks for your detailed response**
> >
> > Thanks for getting back to my comments and providing a detailed response to each of the comments mentioned. I also appreciate your effort to include the comparison against fine-tuning as suggested by other reviewers.
> >
> > I am still unsure about the computational cost statement : *During adaptation, we maintain one estimate of the invariance parameters and update it by gradient. So any “extra calls” just depend on how many gradient steps are allowed, or required to converge*
> >
> > From the code on page 22, it involves making three hyper_network calls apart from the standard forward-pass of the BottleNeck block in ResNet. Are you saying that this is only required during training and not when adapting to a downstream dataset? If it's required for downstream dataset as well, then what I was saying was that it can be computationally equivalent to making three additional forward (and backward if you fine-tune) passes which in turn is equivalent to running three separate jobs with different data augmentations.

---

> > > ### Author Response · Authors · 2022-11-30
> > > **Response to additional questions**
> > >
> > > Thanks for the clarification of your question. We understand the confusion now.
> > >
> > >
> > > The three hypernetworks calls in the forward pass of page 22 are due our strategy of using three smaller hypernetworks to generate the ResNet block-wise, rather than one big hypernetwork to generate the whole ResNet. This is a design decision we made because it was actually faster than using a single big hypernetwork, as we explained in A.1.1.
> > >
> > >
> > > In summary, the three hypernetworks in the forward pass are per-block, and not to be confused with three different augmentations. There is no per-augmentation operation in a forward pass during adaptation.
> > > With regard to pre-training, we also emphasise that only one invariance descriptor is randomly selected at each iteration. The  hypernetworks is then trained to optimize for contrastive loss with corresponding augmentation. This is also completely unrelated to the “three calls” to any augmentation.

---

> > > > ### Comment · Reviewer_SKik · 2022-11-30
> > > > **Thanks for the explanation**
> > > >
> > > > Thanks for explaining it in details, I understand the process much better now. I would increase my score by one point based on our discussion above.

---

### Official Review · Reviewer_oGLZ · 2022-10-24

**Confidence:** 4
**Correctness:** 3
**Technical Novelty And Significance:** 3
**Empirical Novelty And Significance:** 3
**Recommendation:** 6

**Clarity, Quality, Novelty And Reproducibility:**

**Clarity**: the paper is generally pretty clear, although it could definitely be better. My main two issues in that respect are:
    - the explanation of the method (section 3.1) is relatively unclear despite its simplicity. I would highly suggest that the authors add a figure to simply and succinctly show the idea of their method.
    - the authors spend a significant amount of space and experiments on dorsal and ventral invariances, which I find more confusing than useful. First, the names will be confusing for most readers, why not use the standard and self-explanatory "color" vs "geometric"? Second, I don't quite understand the purpose of all those experiments and results. If the point was to have more interpretable results by aggregating over similar augmentations, why not train on all augmentations independently and then simply show the aggregated invariance hyperparameter (eg avg overall dorsal invariance hyperparameter rather than train another model)?

**Quality** my main issue about the paper's quality concerns:
- lack of comparison to the main alternative: full fine-tuning
- lack of discussion about computational efficiency
- figures/tables are generally of poor quality. Eg:
    - text in figures is unreadably small and blurry
    - tables do not follow ICLR requirement of having legends above  the table
    - axes are not always explained (eg figure 1 left) or not self-explanatory without reading the paper (eg the use of "dorsal" and "ventral" instead of standard "geometric" vs "color" augmentations)

**Novelty**: the idea (and instantiation of it) is novel to my knowledge.

**Reproducibility**: seems ok, code snippets and many hyperparameters are provided.

**Strength And Weaknesses:**

**Update**: the authors updated the manuscript to address many of my concerns. In particular, improving the baselines and discussing the computational complexity of their method. I am thus increasing my score 5->6.

-------
**Strenghts**
- **really cool idea** the idea of learning a functional from desired invariances to representations is really nice and I think will be valuable for the community. I actually think that the authors undersell it's usefulness. Eg even when downstream practitioners do know what invariances they want in their downstream tasks, they can now get the desired representations without any retraining. The current paper puts all the emphasis on learning task-specific invariances, but this is only one specific application (and arguably not the best as I say below).
- **simple method** the proposed method is very simple for ViTs, it could thus be easily incorporated in SSL and has the potential of being impactful assuming that the method is indeed useful (see weakness about baselines below). Note: for CNNs it's more complex which will likely hinder it's usefulness.

**Weaknesses**
- (addressed in rebuttal) **Unsuitable baselines: what are the advantages compared to fine-tuning?** my main issue with all the experiments is that I don't think that any of the selected baselines are fair comparisons because those only train the linear probe. Take for example the case of ResNet50, by tuning over the augmentation index i (which directly impacts the weights of your model) you are essentially finetuning all the parameters of your model but comparing to baselines that only train the linear probe. Looking at the Mocov3 results (table 4 vs original paper) it seems that your method performs much worst than full fine-tuning. So what are the advantages of your method compared to full fine-tuning? Is it computational or sample efficiency? If so you should compare to fine-tuning methods under the same computational or sample budget (ie only doing a few passes or doing few shot fine-tuning) [^1]. Your generalization bounds try to give a potential explanation for the gains in terms of better sample efficiency, but this has to be empirically validated given that those bounds are well known to be loose.
- (addressed in rebuttal)  **lack of discussion on scalability** specifically :
    - how efficient are the hypernetworks? what is the increase in the number of parameters and training time?
    - it seems that you restrict yourselves to very few augmentations. Does your method easily scale to many more augmentations?  eg the 14 augmentations set used in autoaugment? Pretraining the model for essentially all typical image transformations would be much more useful as it would allow practitioners to have a fine-grained choice on the desired augmentations.


[^1] Note that the advantage of linear probing vs finetuning is very large in terms of computational gains: because you can prefeaturize your dataset once before training the linear probe, making it orders of magnitude more efficient. In your case, you still have to encode every example at every step. I thus doubt that your hypernetwork is more computationally efficient that fine-tuning a RN50. Even for ViT the gains will likely be minor given that the bottleneck is usually encoding.


**Summary Of The Paper:**

Current SSL representations are pretrained to be invariant to a prespecified fixed set of augmentations. This paper provides a simple way of pretraining representations that can be modified to be invariant to any subset of the pretraining augmentations. They essentially learn a function(al) that maps a set of augmentations to a pretrained encoder invariant to those augmentations. For CNNs this function maps augmentations to weights of the network, while for transformers this simply corresponds to appending the index of augmentations to the input sequence.

They evaluate their representation on transfer learning benchmarks by training both the linear probe and the set of augmentations to be invariant to.  They show significant gains compared to linear probing of standard SSL and  previous baselines that aim to learn representations that are not fully invariant.

**Summary Of The Review:**

The overall idea of the paper is very nice and simple. I believe that it could be useful to practitioners and the research community. My main concern is that the authors only consider a single application for their method (using the data from downstream task to update the parameters) but do not compare or discuss the obvious baseline for this setting: fine-tuning. I thus do not think that it is currently ready for publications. I am happy to improve my score if the authors address this concern, for example by doing one of the following:
- empirically showing advantages compared to fine-tuning (better accuracy, computational efficiency ...)
- explaining why fine-tuning is not the right baseline
- talking and ideally showcasing about other potential applications

---

> ### Author Response · Authors · 2022-11-19
> **Response to Reviewer oGLZ**
>
> Thank you for your feedback and supportive comments that our approach is a “really cool idea”. We aim to comprehensively address your main concerns about baselines and scalability as follows.
>
> **Q1: Unsuitable baselines: what are the advantages compared to fine-tuning?**
>
> Thanks for the comment. Please see our comprehensive discussion of this point in the shared response on comparison to fine-tuning.  In summary, (1) we often beat fine-tuning on average if fine-tuning is restricted to use the same computation budget as ours (Tab2-4). (2) Given that our framework explores an intermediate operating point between linear readout and fine-tuning in terms of number of parameters to update and cost to update, we now present comparisons (Fig 3) of our model vs a range of fine-tuning variants that tune different numbers of layers. The results illustrate how we are systematically strong in the low parameter/low computation regime, even if fine-tuning eventually wins in the high parameter/high computation regime. This shows the interesting intermediate operating point we provide that is different and better to the typical approach of selectively fine-tuning certain layers. (3) Theory: The reviewer is right that deep network bounds are loose (so fine-tuning can only provide vacuous guarantees). However s/he overlooked that our architecture and bound are carefully designed to support non-vacuous bounds. In the expanded theoretical analysis (Sec 5) we show that we can provide non-trivial generalisation error guarantees for CIFAR that are tighter than those of linear models, and certainly tighter than FT which has vacuous bounds.
>
> We believe this comprehensively addresses the reviewer’s criterion (empirically showing advantages compared to fine-tuning) for raising the score.
>
> **Q2a: Scalability of hypernetworks**
>
> The number of parameters in one ResNet50 are 23 million. The number of parameters in our particular implementation of HyperResNet50 is 770 million (approx 33 times more). The key determinant of this is the width of the bottleneck layer of the hypernet. The number of hypernet parameters is dominated by a (Bottleneck Size x Base ResNet size) term. It might well work with a smaller bottleneck, but we didn’t do any architecture search here to see.
> Please note that once the HyperNet is trained during pre-training, the actual adaptation of the hypernetwork to a new task is super efficient (~20k parameters, essentially the same as linear readout) as shown in Fig 3, and exploited in our Theorem. For k>5 invariances, our hypernetwork is more parameter efficient than training 2^k ResNets for each invariance combination, and it is dramatically more computationally efficient.
>
> **Q2b: Number of augmentations.**
>
> For images we used k=2, and k=5 augmentations. For audio we used k=7 augmentations.  It would be no problem to scale to k=14 augmentations as used in supervised AutoAugment, but for now we focused on k=5 in imagery to be consistent with what our self-supervised SimCLR/MoCo baselines used.   Actually, to explicitly model each combination of augmentations, our framework is the only way to scale up, as otherwise training 2^14 unique ResNets is obviously infeasible as discussed above. This fact is a key motivation for our method.
>
> **Q3 Clarity:**
>
> Sorry for the confusion. We took the dorsal/ventral terminology from prior work, but we are happy to change the names to appearance and geometric augmentations to reduce confusion. The reason we did those experiments grouping the augmentations is because (Ericsson et al 2022) already identified these two groups as being functionally related in terms of having similar invariance effect within one family, and being related in terms of their impact of downstream tasks.
> In order to compare our method with a baseline that used one specific ResNet for each invariance option, using these meant we had strong well tuned baselines to compare against. The fact that there were only two groups was convenient because  it allowed us to demonstrate our concept in the simplest case with k=2 options, and crucially meant that we didn’t have the expense of training a huge set of 2^k baselines for k>2 baselines.
>
> We will improve the figures and tables. Thanks for your feedback!

---

> > ### Comment · Reviewer_oGLZ · 2022-11-21
> > **Thank you and remaining concerns**
> >
> > Thank you for your detailed rebuttal, I've read the updated manuscript, which I believe is better than before. My remaining concerns are the following:
> >
> > **Computational requirements**: given that you propose a novel alternative to probing and FT it is really important to have a good comparison of the computational requirements of your method for practitioners to know the practical advantages and disadvantages of your method. In particular, I think that:
> >
> > - you should discuss the computational efficiency (memory and time) of the **pretraining** of your method compared to FT and probing.
> >
> > - you should add the memory requirements for downstream training in figure 3.
> >
> > - probes in figure 3 should be trained on the prefeaturized dataset (I don't think you do it) and the x-axis should then be log-scaled to show the real computational advantage of probing compared to FT and your method. Probing should then be orders of magnitude faster than your method and FT as the bottleneck will be featurizing the dataset once (instead of every epoch).
> >
> > **Use of the cryptic dorsal/ventral terminology**: I've read your answer but I would still strongly suggest changing the ventral/dorsal terminology. The fact that this terminology has been used in one recent paper in the last year is not a reason for favoring it compared to more common and clear/self-explanatory computer vision terminologies such as "geometrical transformation", eg [[1](https://arxiv.org/abs/1708.06020), [2](https://journalofbigdata.springeropen.com/articles/10.1186/s40537-019-0197-0)].
> >
> >
> > ----
> > Answers to your rebuttals:
> >
> > - Q1: Thanks for adding this critical baseline.  In the rebuttals, the authors put much emphasis on Prompt-ViT being an intermediate point between linear readout and fine-tuning. Although I agree in terms of parameters, I think this is somewhat misleading / not important empirically given that you fix the computational budget[^1]. I think that the method should thus not be sold as an "intermediate point" but rather an interesting alternative, which sometimes beats FT with the same computational budget. Concerning the theory thank you for expanding and highlighting that, which is a neat advantage of your method.
> >
> > - Q2a: Thank your answer on the number of parameters. I was asking more generally about computational requirements (time and memory) during pretraining of those networks, which seems much less efficient than standard SSL with a resnet50.

---

> > > ### Author Response · Authors · 2022-11-23
> > > **Response to additional feedback**
> > >
> > > Thanks for your additional feedback.
> > >
> > > **Computational Requirements:**
> > >
> > > **1. Pre-training:** We now provide pre-training memory and time cost in Table 1 below. The pre-training cost is more than a standard RN. However, as for mainstream self-supervision, we see the pre-training cost as an up-front cost that can be amortised across many downstream tasks, and as such not of crucial importance.  We will include it in the camera ready version if accepted.
> > >
> > > **2. Downstream:** We now provide downstream memory cost in Table 2 below. The downstream memory for our method is comparable to that of the full fine-tuning baseline (a bit more for CNN/Hypernet, a bit less for PromptVIT). We will include it in the camera ready version if accepted.
> > >
> > > **Fig 3:** We will update Fig 3(mid,left) to include pre-featurized images for the linear readout baseline in the camera-ready version. Pre-featurising does accelerate the linear readout baseline as correctly identified by the reviewer. However the overall trend that our amortised invariances provide an interesting middle ground between linear readout and fine-tuning remains.  We would like to clear out that in the parameters plot on Fig 3(left) the FT baseline has a constrained budget. In the flops plot on Fig 3(right), the FT baseline has no budget constraint.
> > >
> > > **In summary:** (1) We do have more pre-train clock/memory cost than a vanilla MoCo/RN, (but much less pre-train cost than 2^k vanilla MoCo/RNNs that would be required by a naive scheme to explicitly store all the invariances we learn). (2) We have comparable downstream memory cost to the fine-tuning baselines.
> > >
> > > **Terminology:** Thanks for the reminder. We will revise it throughout to “appearance” and “spatial” for camera-ready version if accepted.
> > >
> > > **Table 1:** Comparison of pretraining cost for amortized MoCo models and baselines. We assume that the models are run serially where one baseline model will require the same memory as $2^k$ baseline model. If we consider parallel training, then memory requirements for $2^k$ baselines increases drastically while clock time remains the same. We show the time taken in GPU days on Tesla V100-SXM2-32GB for 300 pre-training epochs for ResNet50 and ViTs with ImageNet-100 and ImageNet-1k respectively.
> > >
> > > |Method | Ours  |       | Baseline |          |        |         |
> > > |---|-------|-------|----------|----------|--------|---------|
> > > |Model| RN50  | ViT   | 1 RN50   | $2^5$ RN50 | 1 ViT  | $2^5$ ViT |
> > > |GPU (days)| 5.1   | 29.3  | 2.9      | 92.8     | 27.3   | 847.7   |
> > > |Memory(Gb)| 446.4 | 481.6 | 134.4    | 134.4    | 476.8  | 476.8   |
> > >
> > > **Table 2:** Memory requirements (in Mb) for downstream training. FT - f denotes the variants of finetuning where f% of the number of layers/blocks is finetuned. In the case of ResNet50, FT - 50% indicates that the out of 4 layers, the final two layers (3 and 4) are finetuned. For ViTs, FT -  50% is the baseline where block 6 to block 12 of the architecture is finetuned. FT - all denotes the finetuning baseline where the entire model is finetuned.
> > >
> > > |      | Ours  | Linear   | FT-all | FT-75% | FT-50% | FT-25% |
> > > |------|-------|----------|--------|--------|--------|--------|
> > > | RN50 | 13690 | 1452     | 10930  | 7250   | 5238   | 4420   |
> > > | ViT  | 14156 | 1452     | 18916  | 17608  | 15976  | 14468  |

---

> > > > ### Comment · Reviewer_oGLZ · 2022-11-23
> > > > **Score increased**
> > > >
> > > > Thank you for addressing my main concerns,  I increased my score.

---

### Official Review · Reviewer_SHkE · 2022-10-24

**Confidence:** 3
**Correctness:** 2
**Technical Novelty And Significance:** 2
**Empirical Novelty And Significance:** 2
**Recommendation:** 3

**Clarity, Quality, Novelty And Reproducibility:**

The details are described in the previous section. In summary,
- Clarity :: The used notations and some explanations are unclear.
- Quality :: Empirical results are convincing in CNN experiments, but not in ViT ones. Also, this paper does not compare the proposed method with the important baseline, fine-tuning. Theoretical results are not interesting.
- Novelty :: I think The proposed idea is novel.
- Reproducibility :: This paper describes implementation details, but code is not available.


**Strength And Weaknesses:**

Strengths
- The problem of interest, learning augmentation-relevant information, is an important problem in self-supervised learning literature.
- The proposed idea, learning a hyper-network to learn augmentation-specific invariance, is interesting.
- When using CNNs, the proposed method outperforms existing SSL methods that use linear evaluation for downstream tasks.

Weaknesses
- This paper should be compared with fine-tuning, not linear evaluation. This is because the full propagation for whole parameters is necessary to optimize an invariance descriptor. Therefore, its time requirement is the same as fine-tuning. I think fine-tuning can perform better than the proposed method.
- In ViT experiments, the gain over the baseline (MoCo-v3) is marginal.
  - Prompt-ViT outperforms MoCo-v3 on 6 of 10 downstream classification tasks and 4 of 8 few-shot tasks (10 of 18, i.e., 55% in total). This means Prompt-ViT achieves a similar performance to the baseline.
- Theorem 1 is hard to understand and seems to be meaningless.
  - What is $\hat{y}$? What is the definition of the prediction $\hat{y}$?
  - The authors should describe details in the proof of Theorem 1. The sentence "Our result follows from using the union bound to optimise over the choice of feature extractor." is not enough for formal proof.
  - I cannot see the meaning of Theorem 1. The theorem does not tell the benefit of the proposed method. I think the theorem is just a well-known generalization bound (except the third term). The bound can be obtained in any case: fine-tuning, full-invariance learning (e.g., MoCo), etc. IMO, the authors should (theoretically) discuss an advantage over the baseline (linear evaluation using representations obtained from full-invariance learning).
- The writing of this paper can be improved.
  - Confused notation $i$: an invariance descriptor or an instance index (e.g,. see Eq (2)).
  - Confused notation $\mathcal{K}$. In the denominator of Eq (2), $k_{A_i}^i$ is not defined. I suggest using $\sum_{k\in\mathcal{K}}\exp(q_{A_i}\cdot k/\tau)$.
  - Confused notation $\mathcal{D}^t$: pretraining or downstream task dataset.
  - When using MoCo, how to compute the key representations $\mathcal{K}$ for an invariance descriptor $i$?
  - I suggest mentioning that $i$ is a continuous vector when defining $i$.
  - The authors should describe formal details of Hyper-ResNet in the main manuscript.
  - Figure 1 is too blurred. Suggest using high-resolution images.


**Summary Of The Paper:**

This paper aims to learn more generalizable representations. To this end, this paper proposes amortized invariance learning, which learns augmentation-specific invariance by parameterizing the feature extractor by invariance hyperparameters. This paper shows the superiority of amortized invariance learning under various transfer learning benchmarks with both CNNs and ViTs.


**Summary Of The Review:**

Although I feel the methodological novelty, both the empirical and theoretical results are not convincing. Hence, I vote for rejection.

---

> ### Author Response · Authors · 2022-11-19
> **Response to reviewer SHkE**
>
> Thank you for your time and feedback.
>
> **Q1: Comparison to fine-tuning**
>
> Please see our common response to this comment. In summary, the revised Tab 1-4 shows that amortisation can often beat fine-tuning, Fig 3  shows that it offers a favourable accuracy-compute/parameter tradeoff, and crucially the excellent accuracy/parameter tradeoff leads to a much stronger theoretical guarantee than fine-tuning  (revised Sec 5) .
>
> **Q2: Marginal improvement on VIT**
>
> Yes, we agree the improvement for VIT is less. However, while the reviewer has counted the number of wins and found no clear trend, s/he neglects to notice the magnitude of the wins. When we lose to the baseline it is by a small margin, and when we win - especially on regression tasks - it’s often by a huge margin. We have revised the tables to include average results across the various tasks, to make this clear. The average performance is ahead of VIT-linear readout, and similar to VIT-fine-tuned.
>
> **Q3: Theorem 1 unclear**
>
> We have now substantially expanded the exposition of the theoretical analysis part, and added associated results, in order to make the contribution clear. Related theorems can be derived for linear models and fine-tuning, but our key contribution is this particular theorem that is specific to our amortised invariance model. The revision now states the standard linear model & deep net theorems alongside ours in order to better explain the differences and show the significance. In summary: end-to-end fine-tuning bounds are uselessly loose, linear readout bounds can be usefully tight, but linear models have lower accuracy. Our contribution improves on this by both having higher empirical accuracy than linear models, and also a tighter bound than fine-tuning models. See also the common comment on fine-tuning, where we discuss instantiating the bound to show quantitatively that our amortisation framework tightens the bound compared to linear models. This is a significant outcome.
> (Minor Requested Clarifications:: \hat{y} is the prediction of the model for the downstream task, after training as in Eq 4. We changed the notation to add a superscript t to denote task t. We have added more details to the proof to better explain how the union bound is applied to obtain our result. We have added more details to the proof to better explain how the union bound is applied to obtain our result.
>
> **Q4: Writing Points.**
> Thanks. We have tried to clarify all of these.
>
> 1. Confused notation $i$:  In Eq 2, $i$  denotes the invariance descriptor.
> 2. Confused notation $\mathcal{K}$: We realised the mistake in the notation and have corrected it now. $ k^{j}_{\mathbb{A}_i}$ is defined in Eq. 1 is a key corresponding to invariance descriptor $i$.
> 3. Confused notation $\mathcal{D}_t$: $\mathcal{D}_t$ denotes downstream task.
> 4. Keys for MoCo: We maintain a separate queue for each type of invariance encoded in the Hyper-ResNet50 model so that augmented keys corresponding to the same invariance are used for contrastive loss.

---

> > ### Comment · Reviewer_SHkE · 2022-12-02
> > **Response to Authors**
> >
> > Thank you for your efforts and time for this rebuttal. I have read your response, but my concerns are still remaining. Hence, I would like to keep my score.
> >
> > ---
> >
> > **Experimental results**
> >
> > I think comparison with fine-tuning weakens the significance of this work. First, I cannot see a meaningful advantage of the proposed method over fine-tuning. I have read the common response, but I do not agree that the smaller number of learnable parameters is an advantage. Why it is? IMO, in this case, the smaller number of learnable parameters does not provide better performance, faster learning, and so on.
> >
> > ---
> >
> > **Theoretical results**
> >
> > I still think Theorem 1 is still unclear. why the authors do not formally describe the notations and setups? what is $\hat{y}$ and $\phi$? When state a theorem, the authors should describe the theorem formally and thoroughly. Furthermore, I cannot agree that FT has a much larger guaranteed error than linear models. So, I think the guaranteed errors in Table 4 are awkward. Why fine-tuning is much worse than the linear model? IMHO, fine-tuning should have a smaller error since is is more flexible for learning. Also, the theoretical results are related to the invariance learning? Or, any $h(x;i)$ can result in the bound in Theorem 1? The connection between the theory and the method is critical to me.

---

> > > ### Author Response · Authors · 2022-12-03
> > > **Response to additional feedback**
> > >
> > > Thank you for the additional feedback. We have addressed some of your concerns below.
> > >
> > > **Experimental results-** We are not sure why the reviewer "cannot see a meaningful advantage over fine-tuning". To summarise the meaningful advantage:
> > >
> > > 1. In our three of our four quantitative experiments (Tab1,2,3,5) our framework has the highest average rank in accuracy across all datasets. In only table is it second to fine-tuning in average rank.  Our margins are particularly huge for regression tasks. E.g., triple the R^2 performance for few-shot regression on 300W compared to fine-tuning. This alone is a dramatic win, even if someone only ever users our framework for regression.
> > > 2. The results in Fig 3 & similar figures in the appendix show clearly that we do provide substantially faster learning than fine-tuning in terms of clock time. In Fig3(right), the clock time improvement over fine-tuning is ~10-fold.
> > > 3. The smaller number of parameters to learn is directly related to the substantial accuracy improvement that we often obtain. This is because with more parameters to update, the fine-tuning competitor is more at risk of overfitting the downstream training set. Related to the smaller number of parameters, there are substantial theoretical benefits over fine-tuning.  In particular, we have a dramatically tighter generalisation bound than fine-tuning, visible from comparing Thm 5.1 and Thm F.1 (see more below).
> > >
> > > **Theoretical results:**
> > > 1. "Fine-tuning should have a smaller error since it's more flexible for learning". We respectfully point out that this is factually incorrect. Fine-tuning certainly has a strong empirical error on the train set, and might have a strong empirical error on the test set (if the train set was large). However it is a well established fact that can be seen in any seminal reference on learning theory (e.g.,  Shalev-Shwartz and Shai Ben-David, 2014, Understanding Machine Learning: From Theory to Algorithms, Cambridge University Press) that such end-to-end learning provides limited *expected error guarantee* - i.e., limited *guaranteed error on the test set*. This because the same added flexibility that allows zero error on the train set also provides an opportunity for overfitting to it, and thus poor generalisation to the test set.
> > > The above fact is not only well established in the literature, but shown clearly by the trade-off in Thm F.1 (Gouk et al, ICLR'21) that shows the generalisation guarantee for fine-tuning. Here we agree that fine-tuning can set the first term of the RHS to zero, because of its flexibility mentioned by the reviewer. However, the second term of the RHS is inevitably >> 1 for any substantial deep network (note that it's exponential in depth of the network and sub-linear in the size of the dataset), which accounts for the fact that it can overfit to the train set. Thus the overall guaranteed error is still >> 1. The reason that linear models have better guaranteed error  than end-to-end learning is that the corresponding second term in the bound can be < 1 for linear models (Eq 5), while the complexity term is >> 1 for end-to-end models (Thm F.1). This effect dominates the improved empirical error (first term in each bound) provided by fine-tuning over linear models.
> > > 2. "How do the theoretical results relate to invariance learning?". The theoretical result could potentially be applied to some other architectures besides invariance learning, but invariance learning is a very natural fit. Essentially the theory applies to any case where the space of feature extractors can easily be discretised a small set, with the error guarantee being dependent on the size of this set. This doesn't work for fine-tuning, because there are too many parameters to discretise the space of fine-tuned networks into a small set. However, the number of invariance parameters is small enough that it's possible to discretize the set of networks that they correspond to into a small set.
> > >
> > > **Symbols:** $\hat{y}$ is the prediction of our network, stated above eq 4, $\phi$ are parameters of the final linear readout (see Eq 4). Sorry we'll restate these next to the theorem so it's clearer.

---

### Official Review · Reviewer_gbta · 2022-10-27

**Confidence:** 3
**Correctness:** 3
**Technical Novelty And Significance:** 3
**Empirical Novelty And Significance:** 3
**Recommendation:** 8

**Clarity, Quality, Novelty And Reproducibility:**

Overall a well-written paper. Most of the claims are well supported by the analyses and experiments. The quality of the figures (resolution, font, non-vector, etc) needs some improvements.

**Strength And Weaknesses:**

## Strength
1. Thorough analysis to support the claims made in the paper. For example, Fig 1 shows that the final representation indeed adjust adaptively according to the invariance descriptor. Fig 2 also demonstrates that the model can identify the invariance required intrinsically for each downstream task.

1. Strong transfer results to various downstream tasks. This also implicitly confirms the (somewhat intuitive) hypothesis that each downstream task requires different invariance.

1. The authors demonstrate the generalizability of the proposed method on different (1) archs (ResNet and ViT), (2) SSL methods (MoCo and SimCLR), (3) downstream tasks (10 of them spanning across regression and classification tasks), and (4) modalities (vision and audio)

## Weakness

1. It is unclear why training the model with the vanilla contrastive loss in Eq 2 and 3 can encourage the model to produce the final representation conditioned on the invariance descriptor. Without any additional loss or special training techniques, the model can simply ignore the invariance descriptor and degenerate to vanilla SSL models like SimCLR or MoCo. The authors should highlight this part in the Method section.

1. Poor figure quality. For example, they are not vectorized figures and the resolution is not good. The fonts are too small and the shape is distorted (eg. Fig 2)

**Summary Of The Paper:**

This paper proposed to condition the final representation on an invariance descriptor. By doing so, the representation can be adaptively adjusted for target tasks that may require different kinds of invariance.

**Summary Of The Review:**

Overall a good and novel paper. Thorough analyses are conducted to support the claims made in the paper and the final results indeed outperform vanilla SSL methods on various downstream tasks that may potentially require different kinds of invariance.

---

> ### Author Response · Authors · 2022-11-19
> **Response to Reviewer gbta**
>
> Thanks for your feedback and supportive comments!
>
> **Q1: Why does the model produce a representation conditioned on the invariance descriptor, rather than collapse to a default SSL model?**
>
> The invariance descriptors are always paired with the corresponding augmentation during training (mentioned Sec 3.1). This means that if the model did collapse to a default SSL model that ignored the invariance, it would have a higher loss than one that exploited the invariance descriptor to control the invariance. Thus in order to minimise the loss Eq 3 under this paired training condition, it must exploit the invariance descriptors.
>
>
> **Q2: Image quality.**
>
> Sorry for the raster images. We will update the paper to include clearer vector graphics

---

### Author Response · Authors · 2022-11-19
**Response to all Reviewers: Common Question on Fine-Tuning**

Some reviewers mentioned that we should conduct comparison to fine-tuning, rather than linear readout only. We have now updated results to do this, but first we would like to make some remarks contextualize the results:

As we mentioned at the end of Sec 5, our method explores a learning regime that is between fixed feature extractor/linear readout, and conventional fine-tuning. Therefore let us be explicit that neither linear readout nor fine-tuning are fair competitors for our method (just as they are not fair competitors for each other). In terms of parameters fine-tuning has a massive advantage, thanks to learning ~23.5M parameters while linear readout and our amortised framework both learn only ~20K parameters. Meanwhile, fine-tuning and our framework both use more compute than linear readout by relying on backpropagation. We don’t particularly claim that our method will always beat fine-tuning. But we do claim the intermediate regime our method enables is interesting.

**Direct Comparison:** In terms of direct comparison of amortised invariances and fine-tuning (FT), we have now updated all the main tables to include the FT competitor. Following the suggestion of reviewer oGLZ, we constrain the number of FT iterations to use no more than the amount of compute of our model. From the results in the revised Tabs 1-4, we can see that we often but not always beat FT in terms of average performance and average rank across all the datasets. This is a remarkable outcome given the difference in the number of parameters available  (23.5M vs 20K).

**Pareto Analysis:** To emphasise the interesting regime between linear readout and FT in terms of parameters and cost, we compare our model to linear readout and a range of FT variants that fine-tune either one, two, three, or all four ResNet blocks, no with no limitation on compute time for FT. The plots in the revised Fig 3 show the accuracy vs number of parameters to learn (left), and accuracy vs clock time (right) for some example datasets. They also show the pareto front of accuracy vs (parameter/time) efficiency. For 300W, we can see that all the fine-tuning options update vastly more parameters than amortisation and linear readout, with amortisation dominating the vast majority of the pareto front. In terms of compute time, the plots for 300W and CUB show that amortisation costs more than linear readout, but it still dominates a large section of the pareto front of accuracy vs efficiency. On some datasets the fine-tuning models catch up and surpass amortisation, but only after updating massively more parameters and spending substantially more compute.

**Theoretical Analysis:** Besides providing an interesting compute-accuracy trade-of as analysed above, amortisation provides important theoretical benefits. While reviewer oGLZ dismissed our bound as loose, in fact the opposite is true! Conventional deep network bounds (ie: those applicable to FT) are hopelessly loose. But a major contribution of our work is that our novel architecture and associated bound in Thm 5.1 can in fact be tight enough to give non-vacuous guarantees! To demonstrate this, we instantiate our bound for CIFAR10, along with the corresponding bound for linear models and fine-tuning.

|Metric|Baseline-LR|Amortisation|FT|
|:---------------------------------|:------|:--------|:-------|
| guaranteed worst case error | 0.78 |**0.67**| huge |

From these results, we see that: (1) Although the fine-tuning model performs well empirically, it has no theoretical guarantee whatsoever of future generalisation due to accounting for the full complexity of the deep network in the bound, (2) the linear readout method is simple enough to provide a generalisation guarantee, but (3) our amortisation method provides better guarantees than linear readout.

This ability to simultaneously improve empirical performance and theoretical guarantees (as opposed to FT which improves the former and worsens the latter) is a key demonstration of the value of our intermediate regime.

---

### Decision · Program_Chairs · 2023-01-20

**Decision:**

Accept: poster

**Justification For Why Not Higher Score:**

The authors are encouraged to clairfy the details in the paper as suggested by the reviewers.

**Justification For Why Not Lower Score:**

N/A

**Metareview: Summary, Strengths And Weaknesses:**

This paper proposes amortized invariance learning for achieving generalizable representations. Most reviewers agree the idea is interesting and simple at the same time. The results are also demonstrated with different methods and network architectures in SSL. However, there are some concerns of the novelty and the correctness / clarifty of the paper. These have been mostly resolved after the rebuttal. The majority of the reviewers recommended accepting the paper and the AC agrees.

**Note From Pc:**

if the above contains the word "oral" or "spotlight" please see: "oral" presentation means -> notable-top-5% and "spotlight" means -> notable-top-25%. As stated in our emails, we are disassociating presentation type from AC recommendations